# A Rapid Review on the Influence of COVID-19 Lockdown and Quarantine Measures on Modifiable Cardiovascular Risk Factors in the General Population

**DOI:** 10.3390/ijerph18168567

**Published:** 2021-08-13

**Authors:** Alice Freiberg, Melanie Schubert, Karla Romero Starke, Janice Hegewald, Andreas Seidler

**Affiliations:** Faculty of Medicine, Institute and Policlinic of Occupational and Social Medicine, Technische Universität, 01307 Dresden, Germany; melanie.schubert@tu-dresden.de (M.S.); karla.romero_starke@tu-dresden.de (K.R.S.); janice.hegewald@tu-dresden.de (J.H.); andreas.seidler@tu-dresden.de (A.S.)

**Keywords:** COVID-19, lockdown, quarantine, social isolation, cardiovascular, health behaviours, rapid review

## Abstract

Preceding coronavirus outbreaks resulted in social isolation, which in turn is associated with cardiovascular consequences. Whether the current COVID-19 pandemic negatively impacts cardiovascular health is unclear. The aim of the rapid review was to investigate, whether COVID-19 lockdown influences modifiable cardiovascular risk factors (i.e., physical inactivity, sedentary behaviour, smoking, alcohol use, unhealthy diet, obesity, bad blood lipids, and hypertension) in the general population. Medline and EMBASE were searched until March 2021. Title, abstracts, and full texts were screened by one reviewer and 20% by a second reviewer. Only studies using probability sampling were included in order to ensure the representativeness of the target population. Data extraction and critical appraisal were done by one reviewer and double-checked by another reviewer. We identified 32 studies that fulfilled our inclusion criteria. Findings show that physical activity decreased, and sedentary behaviour increased among all age groups during the COVID-19 lockdown. Among adults, alcohol consumption increased, dietary quality worsened, and the amount of food intake increased. Some adults reported weight gain. Studies on children and adolescents were sparse. This rapid review found a high number of epidemiological studies on the impact of COVID-19 lockdown measures on modifiable cardiovascular risk factors, but only a few used probability sampling methods.

## 1. Introduction

According to Gori et al. [1], the COVID-19 pandemic might positively influence the cardiovascular health of the general population: by reduced air pollution, a decreased spread of other infectious diseases, and a temporary decline of traffic-associated noise consequences for the cardiovascular system. However, it might also have a negative impact on cardiovascular health by increasing risk factors of cardiovascular diseases like social isolation, depression, and anxiety, or altered socioeconomic status [1]. Further, health behaviours might be influenced negatively during quarantine—resulting in physical inactivity, unhealthy diet, and thus in an associated weight gain, as well as resulting in increased consumption of tobacco and alcohol [2,3,4,5,6].

Besides the actual absence of social contacts present in social isolation [7], lockdown and quarantine measures during coronavirus outbreaks come along with a feeling of loneliness [8]. Social isolation and loneliness for their parts affect physical and mental health: Apart from leading to depression, increased mortality, and reduced quality of life [9,10], they can affect cardiovascular outcomes. More specifically, in regard to modifiable cardiovascular risk factors (i.e., physical inactivity, sedentary behaviour, smoking, harmful alcohol use, unhealthy diet, obesity, bad blood lipids, and hypertension), general social isolation or loneliness contribute to physical inactivity among (older) adults [11,12,13,14,15,16,17], to an increased tobacco consumption among adolescents and adults [15,16,18,19,20,21,22,23,24], to increased alcohol consumption among (older) adults [15,25,26], and to a rise in prevalence of hypertension in adults [27,28]. Further, social isolation and loneliness in general increase the risk for cardiovascular diseases such as coronary heart disease or stroke [28,29,30].

If these empirical findings on the cardiovascular consequences of social isolation and loneliness, in general, are transferable to these COVID-19 lockdown measures is questionable. The comparability of the exposure variables “social isolation” and “loneliness” with “COVID-19 lockdown measures” is doubtful. The exposure duration of social isolation or the feeling of loneliness of an individual might endure much longer than the time-limited pandemic lockdown of a whole population. In addition, the COVID-19 lockdown might not necessarily result in social isolation or loneliness, as people are still in contact with (family) members of their household, have contact with other persons via social media, are able to meet outside—complying with physical distance measures (depending on their countries’ rules), etc. According to research, it is unclear to date, whether social isolation and a resulting feeling of loneliness were increased during COVID-19 lockdown periods as some publications found no evidence for a rise [31,32], whereas another study reported an increase—at least among 84-year-old adults [33].

Empirical evidence on the impact of COVID-19 lockdown and quarantine measures on cardiovascular health is needed. Several reviews investigated the influence of COVID-19 lockdown measures on a single or a few cardiovascular risk factors [34,35,36,37,38,39,40,41,42,43,44,45,46,47,48,49,50]. Some of these focus on specific age groups like children and adolescents [37,38,39,48], students [40], adults [45,47], or older adults [49]. However, to our knowledge, no review summarizing the impact of COVID-19 pandemic lockdown measures on the entirety of all modifiable cardiovascular risk factors among all age groups has been published yet. Furthermore, all of the previous reviews included studies regardless of the sampling method used, and thus also included studies using non-probabilistic sampling methods. The problem with non-probabilistic samples is that they “may not be representative and findings cannot be safely generalised” to the target population, leading to possible invalid statistical inferences, since participants are selected in a non-random manner [51]. Non-probabilistic sampling methods comprise convenience sampling, judgment sampling, quota sampling, and snowball sampling [51]. In contrast, in studies using probability sampling, “each member of the population has an exactly equal chance of being selected” and are thus more likely to be representative and to determine the true characteristics of a population [51]. Types of probability sampling are full/complete sampling, random sampling, as well as stratified and systematic sampling [51].

Based on this situation, this rapid review aims to investigate whether lockdown and quarantine measures during the COVID-19 pandemic influence known modifiable cardiovascular risk factors in the general population of all age groups, considering only studies with complete (census studies) or probability sampling.

## 2. Materials and Methods

To investigate the objective of this review, the following research question is derived:

“What are the findings of epidemiological observational studies and secondary data studies using representative sampling methods about the influence of COVID-19 lockdown and quarantine measures on modifiable cardiovascular risk factors in healthy persons from the general population of all ages in comparison to no or other forms of quarantine and lockdown measures?”.

The research question is specified by using the PECOS-criteria [52]:Population: general population (all age groups)Exposure: COVID-19 lockdown and quarantine measuresComparison: no quarantine and lockdown measures or different forms of quarantine and lockdown measuresOutcome: modifiable cardiovascular risk factors (i.e., sedentary behaviour, physical inactivity, harmful use of alcohol consumption, smoking, unhealthy diet, obesity, bad blood lipids, and hypertension) [53,54]Study design: epidemiological observational studies (i.e., cohort studies, case-control studies, cross-sectional, studies) using representative sampling methods and secondary data studies

In order to answer the research question, a rapid review was conducted. The standardized procedure is based on recommendations for conducting rapid reviews in the time of COVID-19 by Seidler et al. [55]. The study protocol was published on PROSPERO (CRD42020222405) [56]. To ensure a high reporting quality, we utilized the PRISMA guideline [57].

### 2.1. Inclusion and Exclusion Criteria

We defined the following inclusion and exclusion criteria using the PECOS -scheme (population, exposure, comparison, outcome, and study design) (Table 1) [52].

#### 2.1.1. Population

Healthy persons from the general population of all ages were considered relevant for this paper, which also can be subgroups like students, pupils, and workers. For the latter, it had to be clear, that the effect of lockdown measures on cardiovascular risk factors was investigated—not the effect of work itself. No age restriction was set; meaning that studies investigating children, adolescents, adults, and older adults of the general population were included. Studies on animals received no consideration. For the overall rapid review, patient populations (e.g., obese patients, diabetes patients, patients with cardiovascular disease, or pregnant women) were also relevant, but these are not part of this article. It is planned to publish the results of patient populations in another paper.

#### 2.1.2. Exposure

The exposures of interest were quarantine and lockdown measures during the COVID-19 pandemic. Studies on quarantine and lockdown measures during other pandemics (e.g., SARS, MERS, or Ebola).

#### 2.1.3. Comparison

Studies needed to provide any type of comparison values in order to illustrate the effect of quarantine and lockdown measures, which could be a temporal comparison between a time without and a time with such measures, or a comparison of different forms of such measures (e.g., comparing countries with different lockdown rules). Studies retrospectively asking about a change (e.g., decrease, increase, or no change) in modifiable cardiovascular risk factors since COVID-19 lockdown measures were considered suitable. Studies without any comparison (e.g., studies that only presented prevalence rates without a reference to any change) were excluded.

#### 2.1.4. Outcome

In accordance with information from the World Health Organization and the World Heart Federation, the following variables are reported to be modifiable cardiovascular risk factors and are thus the outcomes of interest: physical inactivity and sedentary behaviour, harmful use of alcohol, tobacco use, unhealthy diet (excessive consumption of (saturated) fat, salt, and sugar, and low intake of fruits and vegetables), obesity, bad blood lipids (hyperlipidaemia, hypercholesterolemia, hypertriglyceridemia), and hypertension [53,54]. Sedentary behaviour is understood as “any waking behaviour characterized by an energy expenditure of less than or equal to 1.5 metabolic equivalents (METs), while in a sitting, reclining, or lying posture”. Additionally, it comprises the use of electronic devices (e.g., television, computer, tablet, phone) [58]. Any form of measurement methods for the outcomes—objective as well as subjective—were suitable for this rapid review. Non-modifiable cardiovascular risk factors (e.g., family history, diabetes, or socioeconomic status) were not considered in this rapid review. We excluded studies considering the impact of COVID-19 lockdown and quarantine measures on hard cardiovascular end points (cardiovascular diseases such as myocardial infarction, stroke, thrombosis, embolism, and arteriosclerosis), since the effect of lockdown measures on cardiovascular diseases is probably obscured by the initially decreased hospitalisation of cases with cardiovascular diseases due to a fear of infection [59,60,61,62,63,64]. Studies on other acute and chronic diseases (e.g., mental disorders, cognitive impairments, musculoskeletal disorders) were also excluded.

#### 2.1.5. Study Design

Epidemiological observational studies (i.e., cohort studies, case-control studies, cross-sectional studies) using representative sampling methods and secondary data studies were of relevance. We base our understanding of representative sampling on the definition by Tyrer and Heyman [51] outlined above. We excluded studies using non-probability sampling. Even though a higher response resembles a higher representativeness, we did not set a minimum value for a response in a study for it to be included. Nevertheless, we evaluated and critically discussed a low response in a study during the risk of bias -assessment. Relevant studies that used study populations of earlier conducted studies, were only included if the original study used probability sampling. Clinical observational studies (case series, case reports), qualitative studies (interviews, focus groups), subjective study types (e.g., editorials, comments, letters), and any type of reviews were not considered relevant. Studies with an abstract only were not considered. Only articles written in English or in German were included.

### 2.2. Inclusion and Exclusion Criteria

The electronic databases MEDLINE (via PubMed) and EMBASE (via Ovid) were searched on March 17, 2021. Search terms for the exposure variables “COVID-19” and “lockdown” as well as for the outcome variables “sedentary behaviour”, “physical inactivity”, “alcohol consumption”, “smoking”, “diet”, “obesity”, “hypertension”, and “bad blood lipids” were used. The search strings were validated by searching a priori defined epidemiological observational studies, which were included in previous reviews on the topic [34,35,41,44,46,47,48,49]. Fifteen of the 16 identified primary studies from these reviews were found with the search strings (accuracy: 93.8%). All search strings were created to emphasize sensitivity, and encompassed medical subject headings and text words. All search strings are displayed in the Appendix A.

A search in the reference lists of all included studies and in topic-related reviews [34,35,36,37,38,39,40,41,42,43,44,45,46,47,48,49,50] supplemented the electronic search. References found through other channels (e.g., expert recommendations or online platforms (e.g., ResearchGate)) were also included if deemed appropriate.

To eliminate all duplicates, the results of the searches were combined in the literature database EndNote.

### 2.3. Study Selection

One reviewer (AF) screened all titles and abstracts, as well as all full texts for eligibility. A second reviewer (MS) screened 20% of all titles and abstracts as well as 20% of randomly chosen full texts in order to check these screening processes. Disagreements were discussed by the two reviewers. In case of a persisting disagreement, two further reviewers (KRS and JH) were involved in the decision process. For the title and abstract screening and full-text screening, a decision guideline outlining the inclusion and exclusion criteria was used. The process of title and abstract screening and full-text screening was piloted by two reviewers (AF and MS) using around fifty titles and abstracts and ten full texts, respectively. The results of this piloting were compared and disagreements were discussed and resolved within the research team. In case no abstract was available, the reference was only excluded, if it was obvious, that the study did not investigate the review topic during title-abstract screening. Otherwise, the full text of the study was retrieved to further check its eligibility. For full-text screening, all excluded studies were documented with the reason for their exclusion. The degree of agreement for a decision between the two reviewers for title-/abstract and full-text screening was determined by calculating Cohens’ Kappa [65].

### 2.4. Data Extraction

Data were extracted in a standardized data extraction sheet by one reviewer (AF). All extractions were double-checked by a second reviewer (MS, KRM, or JH) for accuracy. Disagreements were documented and discussed, if necessary. The following data were extracted: reference (first author name, publication year), methods (study design, study name, country of study, time of study, number of waves, follow-up duration), population (short description, inclusion and exclusion criteria, number of participants invited, number of participants at baseline and follow-up, age, percentage of female participants, response, loss to follow-up), exposure and outcome (description and assessment tool), results (reporting and description of topically relevant results), other information (overall study quality, funding, conflict of interest, methodological strengths and weaknesses). Data extraction was piloted beforehand by two reviewers by extracting three studies independently from each other and comparing and discussing disagreements afterward within the research team.

### 2.5. Critical Appraisal

The methodological quality of included studies was assessed by one reviewer (AF) and double-checked by a second reviewer (MS, KRS, or JH), using a risk of bias -assessment tool following Ijaz et al. [66] and Kuijer et al. [67]. Risk of bias in nine study domains was judged as “low”, “high”, or “unclear”, whereby six domains were major domains (1. recruitment procedure and follow-up (in cohort studies), 2. exposure definition and measurement, 3. outcome source and validation, 4. confounding, 5. analysis method, and 6. chronology) and three domains minor domains (7. blinding of assessors, 8. funding, and 9. conflict of interest). The overall risk of bias evaluation of a study was based on the assessment of the major domains. If all six major domains were judged to be of low risk of bias, the overall risk of bias of a study was low. Otherwise, the overall risk of bias of a study was high. Quality assessment was piloted using three studies.

### 2.6. Data Synthesis

Study results were summarized descriptively and in summary tables, sorted by outcome parameters (i.e., sedentary behaviour, physical inactivity, harmful use of alcohol consumption, smoking, unhealthy diet, obesity, bad blood lipids, and hypertension), age groups (children, adolescents, adults, older adults), and specific population groups (e.g., students, members of sports associations, users of fitness tracking apps).

We extracted the following effect measures if reported in the studies: prevalence and incidence of outcome changes, prevalence and incidence of at least two different time points (with versus without exposure, ideally stating a *p*-value), effect measures for the relative risk of an outcome (e.g., incidence rate ratio, relative risk, hazard ratio, odds ratio, prevalence ratio), and effect measures for continuous outcomes (i.e., mean differences).

## 3. Results

### 3.1. Results of the Literature Search

The database search in PubMed and EMBASE yielded 3760 hits. After duplicate removal, 2844 titles and abstracts (including three relevant hits found through hand searches) were screened, of which 548 references were included in the full-text screening. Overall, 33 full texts were eligible for inclusion: thirty identified by database searches and three by hand searching reference lists.

The 515 full texts excluded are listed in the Appendix A. Thirty-six papers investigated the influence of lockdown measures on cardiovascular risk factors among patient cohorts and are thus not within the scope of this paper. Nearly half of the full texts screened (*n* = 256) were topically relevant (regarding healthy persons and/or patient cohorts), but used one or more forms of non-probabilistic sampling to recruit participants and thus were irrelevant for this review. Most of these studies launched their study via social media (e.g., Facebook, Instagram, or WhatsApp), used mailing lists or panel registers hosted the study on websites, or promoted it with flyers, newspapers, etc. Some studies were excluded because participants were recruited from another ongoing study that originally used non-probabilistic sampling. Thirty-six studies seemed to be relevant for the review purpose, but information on the recruitment procedure was missing, and corresponding authors did not respond to e-mails requesting information. Seventy-one full texts were excluded due to their study design (e.g., editorials, comments, narrative reviews). Forty-five full texts could have been of interest for the review purpose, but reference values to judge the lockdown effect were missing. Other reasons for exclusion of a full text were: irrelevant exposure (e.g., COVID-19 pandemic in general) (*n* = 16), publication language other than English or German (*n* = 15), double publication (*n* = 1), and irrelevant population (*n* = 1). In addition, six full texts were not accessible, despite extensive efforts made by our librarian. The degree of agreement between reviewers for title/abstract screening is substantial (Cohens’ Kappa: 0.61 [65]), and for the full-text screening, it was moderate (Cohens’ Kappa: 0.41 [65]) according to Landis and Koch [68]. The results of the literature search are summarized in Figure 1.

### 3.2. Study Characteristics

Thirty-two studies (from 33 publications) investigated whether modifiable cardiovascular risk factors changed during COVID-19 confinement measures [32,33,69,70,71,72,73,74,75,76,77,78,79,80,81,82,83,84,85,86,87,88,89,90,91,92,93,94,95,96,97,98,99]. Most studies were cross-sectional studies (*n* = 19), nine were cohort studies, and four were prospective secondary data analyses. Most studies were conducted in Europe (*n* = 17), thereof five in the United Kingdom, two each in France, Norway, Italy, Spain, and Turkey, and one each in Croatia and Germany. Nine studies were executed in North America (US: *n* = 6, Canada: *n* = 3), four in Asia (Japan: *n* = 2, China: *n* = 1, United Arab Emirates: *n* = 1), and one each in Australia and Brazil.

The surveys of the cross-sectional studies were carried out in March (*n* = 1), March/April (*n* = 3), April (*n* = 1), April/May (*n* = 4), May (*n* = 5), May/June (*n* = 1), June (*n* = 1), and August (*n* = 1) 2020. Two cross-sectional studies did not report the time of their survey. Most cohort studies conducted the survey during lockdowns (exposure) in March/April/May 2020. Assessments periods before lockdown measures (pre-exposure time) varied widely, from baseline assessments carried out in 2015, 2016, 2017, 2018, or 2019. Some studies even undertook baseline investigations at the beginning of 2020. These different time points of baseline assessments led to varying follow-up durations (4 months–5 years). Only one cohort study also made an outcome measurement post-lockdown additionally to measurement during lockdown [95]. The four (prospective) secondary data analyses based their measurements on continuous data collection via movement tracking systems, whereby two calculated mean values for the time before and during lockdown [81,94] and two presented linear trends (trajectories) of outcomes before and during lockdown measures [98,99].

Different sampling approaches were used. Seven studies recruited adults from the general population via community registers [69,75,78,79,82,90,91]. Six studies enrolled adult participants from previous, ongoing studies [32,33,77,80,83,85]. Participants in one study (two articles) were adult twins from the Washington State Twin registry [72,73]. Five studies invited all or a random sample of students from universities [74,76,84,88,92]. Another study only used data from students who used a university-intern movement tracking app [98]. Two studies used data from all registered adult users of a movement tracking app [81,94]. One study recruited all adults of a Norwegian sports association [70]. Another study enrolled all older adults from a continuity care retirement community [99]. One study recruited all middle-aged and older adults undergoing annual physical check-ups who used the WeChat app [96]. The five studies on children and adolescents either enrolled participants at school [87,89,95] or via community registers [86,93].

The majority of studies investigated adults (≥18 years) (*n* = 27), and five studies investigated children and adolescents. In studies on adults, the mean (or median) age ranged from 49.0 to 64.6 years (if reported) [69,70,71,72,73,77,82,96]. Two studies in the elderly reported a mean age of 67 years [75] and 74.5 years [91]. Four studies researched participants with a very narrow age span, as they were drawn from specific birth year cohorts (in 1936 [33] and 1970 [80]) or class years [83,85]. Seven studies gave no information on the mean or median age of participating (older) adults [32,78,79,81,90,94,99]. The mean (or median) age of the subgroup of students ranged between 20.0 and 29.9 years [74,76,84,88,92,97,98]. Of the five studies investigating children and adolescents, three stated a mean age between 9.0 and 12.1 years [87,89,93], and two an age range of 5–17 years [86] and 15–18 years [95], respectively.

With regard to the proportion of females and males, in studies on adults, an almost equal distribution—meaning a percentage of females of 45–55—was found in six studies [32,33,75,80,82,91]. Twelve studies had a higher percentage of females [69,72,73,77,79,81,83,85,90,94,96,99], and only one study (two articles) had a higher percentage of males [70,71]. All studies using students as a population had a higher percentage of females, ranging between 60.0 and 80.0 percent [74,76,84,88,92,97,98]. Four studies on children and adolescents which gave information on gender distribution, illustrated a nearly equal distribution of boys and girls [86,87,89,93]. Overall, only two studies did not state values of gender distribution [78,95].

In regard to lockdown measures, eleven studies specified concrete general actions taken, e.g., “stay-at-home” orders, the requirement to work from home, closure of cultural and sports facilities and other non-essential businesses, closure of educational institutions, quarantine/isolation requirements in case (of a suspicion) of COVID-19 infection, travel restrictions, social distancing rules, prohibition of gatherings (social and public), etc. [32,69,75,78,79,80,82,83,92,94]. Four studies on children and adolescents focussed on lockdown measures for this age group [86,87,89,93]. The exposure of interest in four studies that investigated the subgroup of students was the closure of university campuses [74,84,88,97]. Tornaghi et al. [95] reported only sport-specific lockdown regulations. Yamada et al. [99] addressed regulations of the care facility where the study was conducted. Twelve studies gave no examples of country-specific lockdown rules [33,70,71,72,73,76,77,81,85,90,91,96]. Two studies further measured self-reported individual lockdown measures: Alpers et al. [69] used being placed in quarantine and being temporarily laid off, or being in the home office; and Crochemore-Silva et al. [79] used practicing social distancing (i.e., staying at home and avoiding contact with other people) or an activity routine (ranging from staying at home all the time to going out every day to work or to perform other regular activities).

The studies investigated the following factors, which influence the cardiovascular system: physical activity (*n* = 21), alcohol consumption (*n* = 8), sedentary behaviour (*n* = 8), weight/body-mass-index (*n* = 6), eating behaviour (*n* = 5), smoking (*n* = 5) and antihypertensive/lipid-lowering/hypoglycaemic medication (*n* = 1).

Table 2 gives an overview of the study characteristics of each study. More detailed information on study characteristics is outlined in the Appendix A.

### 3.3. Results of the Risk of Bias-Assessment

The overall risk of bias was evaluated to be low in only two of the 32 included studies [32,87], but for the prospective cohort study of Niedzwiedz et al. [32] this applies only for the outcome “alcohol consumption”, which was measured with a validated instrument. All other thirty studies were judged to be of an overall high risk of bias, mainly due to the lacking possibility of cross-sectional studies to show a true temporal relationship between exposure and outcome, and/or due to missing or low response, and/or due to high loss to follow-up values.

The major domain “Recruitment procedure and follow-up (in cohort studies)” had a high risk of bias in 21 studies because response (<50%) was too low and/or loss to follow-up (>20%) too high and no non-responder or drop-out analysis was conducted. Recruitment procedures of all included studies were judged to have a low risk of bias since it was a requirement for inclusion to have used complete or any forms of probability sampling (see Section 2.1).

All studies were evaluated to have a low risk of bias for the major domain “Exposure definition and measurement” because all study participants experienced COVID-19 lockdown measures.

Most studies (*n* = 18) received a high-risk evaluation of the major domain “Outcome source and validation”, either because outcomes were measured with only one or few unvalidated self-reported questions or in case of movement tracking, it is assumed that app data objectively measured walking levels (steps), but did not measure other types of physical activities and thus may have underestimated the actual physical activity level of an individual. In seven studies, outcomes were measured with validated instruments, leading to a low risk of bias judgement. Seven studies had different risk of bias evaluations—namely low risk of bias as well as high risk of bias—depending on the outcome since some outcomes were determined with valid instruments while others were measured with not-validated single questions. The differing risk of bias assessment for this category led to two different overall risks of bias evaluations of the study by Niedzwiedz et al. [32].

Fifteen studies had a low risk of bias in the major domain “Confounding and effect modification” since these took account of the variables “sex” and “age” during data analysis (e.g., by stratification, adjustment, or interaction analysis). In some studies, the age span of participants was very narrow, so that we assumed that this variable had no effect on results.

The majority of studies (*n* = 20) statistically compared (prospectively or retrospectively measured) outcome values prior and during COVID-19 lockdown measures with adequate statistical tests, and thus they were judged to have a low risk of bias for the major domain “Analysis method”. Studies that evaluated changes of outcomes since lockdowns by self-report were classified as “high risk”.

Only the nine prospective cohort studies and the four prospective secondary data analysis studies received a low risk of bias assessment for the major domain “Chronology”, since the exposure preceded the outcome, and thus a temporal relation might be established. Nevertheless, for three cohort studies, the follow-up duration of ≥2 years might have been too long in order to investigate the real lockdown effect, as outcomes may have been changed due to other reasons over such a long time [32,80,93]. Even though the cross-sectional studies asked about a change of outcome measures since COVID-19 lockdowns or retrospectively gathered data, a recall bias could not be ruled out.

The minor category “Blinding of assessors” was assessed with a low risk of bias for all studies since researchers did not have direct contact with participants. Therefore the knowledge of the exposure status of a person should not have influenced the results. Further, in all studies, all participants were affected by lockdown measures.

The minor domain “Funding” was judged to be of low risk in 27 studies, either because a study received no external financial support or because the organizations supporting the study clearly did not affect the study results. Only five studies gave no information regarding funding, resulting in an “unclear” assessment [78,79,84,89,99].

In five studies, a statement about “Conflict of interests” was missing [78,79,80,97,99]. All other studies declared (or were assessed) to have no conflict of interest.

Study-specific risk of bias -assessment results are reported in Table 3.

### 3.4. Results from the Included Studies

The results of prospective studies, which are able to show a temporal association—namely of cohort studies and secondary data analysis—are reported descriptively and in summary tables for each cardiovascular risk factor (Table 4, Table 5, Table 6, Table 7, Table 8, Table 9 and Table 10). The results of cross-sectional studies, which are not able to outline “true” temporality and which are prone to recall bias, are shown in the summary tables only. Detailed study findings are outlined in the Appendix A.

#### 3.4.1. Physical Activity

One cohort study that addressed adults found a statistically significant decrease in physical activity levels during COVID-19 lockdown [33]. There were four prospective studies on adults which used movement tracking data. One Canadian study indicated that moderate-to-vigorous physical activity was only significantly reduced in the first, but not in the sixth week of confinement during the pandemic among adults [81]. Another study from Australia confirmed these findings and further showed that the average number of steps per day significantly increased after the relaxation of COVID-19 restriction measures [94]. A Chinese study illustrated that mean daily steps dropped from 8097 to 5440 during physical distancing measures among middle-aged and older adults [96]. Another study from Japan, which measured walking distance of physically independent residents of a care retirement community with a beacon transmitter, outlined a gradual daily decrease at a rate of 0.5% after the announcement of the cancellation of all upcoming in-facility events and exhibitions and the closure of some common facilities as a precautionary measure on 24 February 2020 [99]. The Japanese state of emergency declaration from 7 April 2020 with the order to stay at home had a further significant acute impact on daily walking distance, marked by a 20.3% decrease [99].

One prospective cohort study Savage et al. [92] focussed on students and found that physical activity decreased during the first five weeks of lockdown among UK students (statistically significant at *p* < 0.01)

Three cohort studies investigated the influence of lockdown in general and of school closures in particular, on physical activity levels of children and adolescents [87,93,95]. In the cohort study of Medrano et al. [87], the time spent physically active among 8–16 years old pupils significantly decreased from 154 min/day in late 2019 to 63 min/day during home confinement (*p* < 0.001). Further, around 95% reported having worsened their physical activity lifestyle [87]. Schmidt et al. [93] found a decrease of 11 min per day in the total amount of sports (statistically significant at *p* < 0.01), but also found a 36-min increase in the daily time spent with habitual activities (i.e., playing outside, walking and cycling, gardening, housework) among children and adolescents from Germany during the COVID-19 lockdown compared to the time in August 2018. Furthermore, there was a significant increase of days being active for more than 60 min with moderate to vigorous intensity per week (*p* < 0.01) in this cohort [93]. Among adolescents in a cohort study from Italy, the proportion of those being moderately active decreased during lockdown from 66.3% in January 2020 to 53.6% in April 2020 but increased shortly after relaxation of lockdown rules to 61.7% in May 2020 [95]. The proportion of physically inactive adolescents decreased from January to April from 17.8% to 25.8% but stabilised at the initial level post-lockdown (18.5%) [95]. In contrast, the proportion of adolescents intensively active increased during lockdown from 15.8% to 19.8% [95].

**Table 4 ijerph-18-08567-t004:** Results on physical activity. * adjusted for year, age group, gender, ethnicity, period and period × age group interaction, ** adjusted for year, age group, gender, ethnicity, period and period × gender interaction.

Reference (Study Design)	Country	Population (Sample Size)	Results			
Children and Adolescents						
Medrano et al., 2020 [87] (Cohort study)	Spain	School children aged 8–16 years (baseline: *n* = 281, follow-up: *n* = 113)	**Change since lockdown**			
				**T1 (before lockdown)**	**T2 (during lockdown)**	***p***
				**(M SD))**	**(M SD))**	
			**Physical activity (minutes/day)**	154 (40)	63 (39)	<0.001
			**Change since lockdown**			
				**Prevalence (%)**		
			**Worsening of physical activity**	95.2		
Schmidt et al., 2020 [93] (Cohort study)	Germany	Children and adolescents (baseline: *n* = 2722, follow-up: *n* = 1711)	**Change since lockdown**			
				**Baseline (%)**	**Follow-up (%)**	***p***
			**Days active (days/week) for more than 60 min with moderate to vigorous intensity**	4.3 (1.8)	4.7 (2.0)	<0.01
			**Physical activity guideline adherence**	19.1	30.1	<0.01
			**Total amount of (organized and non-organized) sports (minutes per day)**	34.9 (26.0)	24.3 (36.2)	< 0.01
			**Total amount of (organized and non-organized) sports (minutes per day)**	34.9 (26.0)	24.3 (36.2)	< 0.01
Tornaghi et al., 2020 [95] (Cohort study)	Italy	Adolescents (15–18 years) (baseline: *n* = 1568, follow-up: *n* = 1568)	**Change since lockdown**			
				**Pre-lockdown**	**During lockdown**	**Post-lockdown**
				**(*n* (%))**	**(*n* (%))**	**(*n* (%))**
			**Physically inactive**	154 (17.8)	102 (25.8)	53 (18.5)
			**Moderate activity**	573 (66.3)	214 (53.6)	177 (61.7)
			**Intense activity**	137 (15.8)	79 (19.8)	57 (19.9)
			**Change since lockdown**			
				**Pre-lockdown**	**During lockdown**	**Post-lockdown**
				**(M (SD))**	**(M (SD))**	**(M (SD))**
			Physical activity (minutes/week)	1676 (21)	n.r.	1775 (34)
			- statistically significant difference in physical activity measured as MET-min/week, absolute, or categorical physical activity levels (3 × 3 ANOVA): higher physical activity during and after lockdown than before			
McCormack et al., 2020 [86] (Cross-sectional study)	Canada	Children aged 5–17 years (*n* = 328)	**Change since lockdown**			
			**Prevalence (*n*%)**			
			**Physical activity at home**			
			Increased	48.8		
			No change	32.9		
			Decreased	18.3		
			**Physical activity outdoors**			
			Increased	38.7		
			No change	22.3		
			Decreased	39		
			**Playing at a park**			
			Increased	15.5		
			No change	31.7		
			Decreased	52.7		
			**Playing at other public places**			
			Increased	9.5		
			No change	36.9		
			Decreased	53.7		
**ADULTS**						
Savage et al., 2020 [92] (Cohort study)	United Kingdom	Students (baseline: *n* = 1477, follow-up: *n* = 214)	**Change since lockdown**			
				***p***	**Cohens’ d**	
			**Moderate to vigorous physical activity levels**	<0.01 **	0.12	
Wickersham et al., 2021 [98] (Prospective secondary data analysis)	United Kingdom	Students who had enrolled in the remote measurement techno-locy (RMT) King’s Move Physical Activity (PA) tracker app (*n* = 736)	**Change since lockdown**			
			**Steps/week**	**IRR (95% CI)**	***p***	
			Linear effect	1.00 (0.97–1.03)	0.984	
			Quadratic effect	1.00 (1.00–1.01)	0.047	
Barkley et al., 2020 [74] (Cross-sectional study)	United States	Students (baseline: *n* = 184)	**Change since campus closure**			
				**Pre-campus closure (M (SD))**	**Post-campus closure (M (SD))**	
			**Mild physical activity**			
			Undergraduate students	16.3 (22.6)	10.8 (12.9)	
			Graduate students	12.0 (22.4)	11.2 (11.7)	
			**Moderate physical activity**			
			Undergraduate students	15.0 (15.7)	12.9 (12.4)	
			Graduate students	17.1 (36.9)	16.6 (19.7)	
			**Strenuous physical activity**			
			Undergraduate students	16.0 (22.1)	14.0 (17.9)	
			Graduate students	19.1 (32.9)	21.0 (33.7)	
			**Total physical activity**			
			Undergraduate students	47.2 (40.2)	37.7 (30.7)	
			Graduate students	48.2 (75.2)	48.7 (58.8)	
Özden and Kilic, 2021 [88] (Cross-sectional study)	Turkey	Nursing students (*n* = 1011)	**Change since lockdown**			
				**Before COVID-19 outbreak (%)**	**During lockdown (%)**	
			**Regular exercise every day**	32.6	43.3	
Karuc et al., 2020 [83] (Cross-sectional study)	Kroatia	Young adults (*n* = 91)	**Change since lockdown**			
			**Physical activity**	**Prevalence (%)**		
			**Women**			
			No change	25		
			Increase	19		
			Decrease	56		
			**Men**			
			No change	31		
			Increase	19		
			Decrease	50		
			Change since lockdown			
			**Moderate-to-vigorous physical activity (minutes/day)**	**Pre-restrictions**	**Post-restrictions**	***p***
				(Median (IQR))	(Median (IQR))	
			**Women**	120.0 (227.1)	64.3 (75.0)	>0.0001
			**Men**	135.0 (127.5)	85.7 (56.8)	0.006
Di Sebastiano et al., 2020 [81] (Prospective secondary data analysis)	Canada	Adults (≥18 years) using a physical activity tracking app (baseline: *n* = 2338, follow-up: 2388 (only complete data sets were used))	**Change since lockdown**			
				**4 weeks prior physical distancing (M (SE))**	**1 weeks after beginning of physical distancing (M (SE))**	***p***
			**Moderate-to-vigorous physical activity (minutes)**	194.2 (5.2)	176.7 (5.0)	<0.001
			**Light physical activity (minutes)**	1000.5 (17.0)	874.1 (15.6)	<0.001
			**Steps**	48,625 (745)	43,395 (705)	<0.001
			**Change since lockdown**			
				**4 weeks prior physical distancing (M (SE))**	**6 weeks after beginning of physical distancing (M (SE))**	***p***
			**Moderate-to-vigorous physical activity (minutes)**	194.2 (5.2)	204.4 (5.4)	0.498
			**Light physical activity (minutes)**	1000.5 (17.0)	732.0 (14.3)	<0.001
			**Steps**	48,625 (745)	41,946 (763)	<0.001
To et al., 2021 [94] (Prospective secondary data analysis)	Australia	Adults using a physical activity tracking app (baseline: *n* = 60,560, follow-up: 2388 (only complete data sets were used))	**Change since lockdown**			
				**Before lockdown**	**After lockdown**	***p***
			**7-day average of steps per day**	9500	9175	<0.001
			**30-day average of steps per day**	9684	9199	<0.001
Wang et al., 2020 [96] (Cohort study)	China	Middle-aged and older adults (≥40 years) using a physical activity tracking app (baseline: *n* = 4145, follow-up: 3544)	**Change since lockdown**			
				**Comparison 2019 with lockdown (mean difference (95% CI))**	**Comparison early 2020 with lockdown (mean difference (95% CI))**	
			**Number of daily steps**	−413 (−501–(−325))	−2672 (−2763–(−2582))	
Crochemore-Silva et al., 2020 [79] (Cross-sectional study)	Brazil	Adults (*n* = 377)	**Change in leisure time physical activity according to level of social distancing**			
			**Level of social distancing**	**Engaging in physical activity (%)**	***p***	
			Very little	~20	0.023	
			Little	Not reported (~21 *)		
			Average	37.7		
			A lot	Not reported (~25 *)		
			Virtually isolated	~20		
Duncan et al., 2020 [73] (Cross-sectional study)	United States	Adult twins (*n* = 3971)	**Change since lockdown**			
			**Physical activity**	**Prevalence (%)**		
			Decreased a lot	15.1		
			Decreased somewhat	28.7		
			No change	26.4		
			Increased a lot	5.2		
			Increased somewhat	21.2		
Okely et al., 2020 [33] (Cohort study)	Scotland	Older adults (born in 1936) (baseline: *n* not reported, follow-up: *n* = 137)	**Change since lockdown**			
				**Baseline (2017–2019) (*n* (%))**	**Follow-up (2020) (*n* (%))**	***p***
			**Only household chores**	14 (10.2)	26 (19.0)	0.012
			**Outdoor activities 1–2×/week**	28 (20.4)	23 (16.8)	
			**Outdoor activities >2×/week**	67 (48.9)	74 (54.0)	
			**Moderate exercise 1–2×/week**	19 (13.9)	4 (2.9	
			**Moderate exercise >2×/week**	6 (4.4)	10 (7.3)	
			**Keep-fit/heavy exercise several times/week**	3 (2.2)	0 (0.0)	
Yamada et al., 2020 [99] (Cohort study)	Japan	Physically independent residents, living in a continuing care retirement community (baseline: *n* = 114, follow-up: *n* = 114)	- after the continuing care retirement community announcement (24 February 2020) until the state of emergency declaration (7 April 2020), walking distance gradually decreased at a rate of 0.5% [5.4 m/day (95% CI: −10.4–(−0.4))] - the state of emergency declaration had a further significant acute impact on the daily walking distance by a 20.3% decrease [−186.8 m (95% CI: −333.0–(−40.6))]			
Berard et al., 2021 [75] (Cross-sectional study)	France	Older adults (aged ≥50 years) (*n* = 536)	**Change since lockdown**			
				**Prevalence (*n* (%))**		
			**Decreased physical activity**	194 (36.2)		
Sasaki et al., 2021 [91] (Cross-sectional study)	Japan	Older adults (60–95 years) (baseline: *n* = 2008)	**Change since lockdown**			
				**Before restrictions**	**After restrictions**	*p*
				**(M (SD))**	**(M (SD))**	
			**Vigorous physical activity (MET)**			
			Men	1690.6 (2668.8)	1604.8 (2598.2)	0.035
			Women	742.5 (1701.3)	717.5 (1738.0)	0.4
			**Moderate physical activity (MET)**			
			Men	1064.7 (1332.8)	1002.6 (1306.4)	0.0024
			Women	712.5 (1062.7)	644.4 (1005.1)	0.0022
			**Walking (MET)**			
			Men	922.9 (1035.5)	877.4 (1028.9)	0.0054
			Women	717.2 (899.6)	647.2 (870.5)	<0.001
			**Total physical activity (MET)**			
			Men	3678.2 (4163.1)	3484.8 (4112.3)	0.0024
			Women	2172.1 (2873.2)	2009.2 (2876.6)	<0.001

#### 3.4.2. Sedentary Behaviour

Two prospective cohort studies—one from Spain [87] and one from Germany [93]—investigated the effect of COVID-19 lockdown measures on sedentary behaviours among children and adolescents and showed a significant increase in screen time among this age group since school closures, whereby Schmidt et al. [93] described an increase of around one hour per day during lockdown compared to August 2018. Such findings were also found for student populations: a prospective cohort study from the UK [92] described a statistically significant increase in sedentary time among participating students.

**Table 5 ijerph-18-08567-t005:** Results on sedentary behaviour. * adjusted for year, age group, gender, ethnicity, period and period × age group interaction.

Reference (Study Design)	Country	Population (Sample Size)	Results			
Children and Adolescents						
Medrano et al., 2020 [87] (Cohort study)	Spain	School children aged 8–16 years (baseline: *n* = 281, follow-up: *n* = 113)	**Change since lockdown**			
				**T1 (before lockdown)**	**T2 (during lockdown)**	***p***
				**(M SD))**	**(M SD))**	
			**Screen time (hours/day)**	4.3 (2.4)	6.1 (2.4)	<0.001
			**TV time ≥2 h/day (N, %)**	3 (2.8)	14 (13.2)	0.005
			**Videogame time ≥2 h/day (N, %)**	6 (5.7)	7 (6.6)	0.775
			**Computer (no homework) ≥2 h/day (N, %)**	1 (0.9)	0 (0.0)	0.316
			**Total mobile-phone ≥2 h/day (N, %)**	4 (3.8)	20 (18.9)	0.001
			**Total screen time ≥2 ≥2 h/day (N, %)**	70 (66.0)	93 (87.7)	<0.001
			**Change since lockdown**			
				**Prevalence (%)**		
			**Worsening of screen time**	68.9		
Schmidt et al., 2020 [93] (Cohort study)	Germany	Children and adolescents (baseline: *n* = 2722, follow-up: *n* = 1711)	**Change since lockdown**			
				**Baseline (%)**	**Follow-up (%)**	***p***
			**Screen time guideline adherence**	60.9	37.6	<0.01
			**Recreational screen time (TV, gaming, recreational internet) (minutes per day**	133.3 (123.1)	194.5 (141.3)	<0.01
McCormack et al., 2020 [86] (Cross-sectional study)	Canada	Children aged 5–17 years (*n* = 328)	**Change since lockdown**			
				**Prevalence (*n*%)**		
			**Watching TV**			
			Increased	58.8		
			No change	38.4		
			Decreased	2.7		
			**Playing video games**			
			Increased	56.4		
			No change	40.9		
			Decreased	2.7		
			**Using screen-based devices**			
			Increased	75.9		
			No change	22		
			Decreased	2.1		
Ozturk Eyimaya and Yalçin Irmak, 2020 [89] (Cross-sectional study)	Turkey	Children aged 6–13 years (*n* = 1155)	**Change since lockdown**			
			**Screen time**	**Prevalence (*n*%)**		
			Increase	71.7		
			Decrease	6.1		
			No change	23.2		
**ADULTS**						
Savage et al., 2020 [92] (Cohort study)	United Kingdom	Students (baseline: *n* = 1477, follow-up: *n* = 214)	**Change since lockdown**			
				***p***	**Cohens’ d**	
			**Time spent in sedentary behaviour on a typical day in the last month**	<0.0001 *	0.78	
Barkley et al., 2020 [74] (Cross-sectional study)	United States	Students (baseline: *n* = 184)	**Change since campus closure**			
			**Sedentary behaviour (minutes/week)**	**Pre-campus closure (M (SD))**	**Post-campus closure (M (SD))**	
			Undergraduate students	3089.2 (1455.4)	3681.0 (1600.3)	
			Graduate students	3129.1 (1329.7)	3696.4 (1566.5)	
			- statistically significant (*p* = 0.003) main effect of time for sedentary behaviour			
Colley et al., 2020 [78] (Cross-sectional study)	Canada	Adults (baseline: *n* = 4524)	**Increase since lockdown**			
			**Watching TV**	**Prevalence (% (95% CI))**		
			Men	59.8 (56.3–63.2)		
			Women	66.0 (63.2–68.6)		
Sasaki et al., 2021 [91] (Cross-sectional study)	Japan	Older adults (60–95 years) (baseline: *n* = 2008)	**Change since lockdown**			
			**Sitting time (minutes/day)**	**Before restrictions**	**After restrictions**	***p***
				**(M (SD))**	**(M (SD))**	
			Men	273.4 (203.4)	287.7 (204.1)	<0.001
			Women	243.7 (181.5)	267.8 (191.6)	<0.001

#### 3.4.3. Alcohol Consumption

One UK cohort study on alcohol consumption among adults found an increased relative risk for binge drinking and alcohol frequency during lockdown compared to the years 2017–2019 (relative risk: 1.48 (95% CI:1.27–1.73) and relative risk: 1.38 (95% CI: 1.26–1.51), respectively) and a decreased risk for heavy drinking (relative risk: 0.46 (95% CI: 0.32–0.66)) [32]. Contrary to this finding, another cohort study from the UK found a significant increase in the prevalence of high-risk drinking among adults during lockdown compared to the period from 2016 to 2018 of 5.2 percentage points [80].

**Table 6 ijerph-18-08567-t006:** Results on alcohol consumption.

Reference (Study Design)	Country	Population (Sample Size)	Results				
Adults							
Niedzwiedz et al., 2020 [32] (Cohort study)	United Kingdom	Adults (baseline: *n* = 27,141, analysed at follow-up: *n* = 9748)	**Association between lockdown and …**				
				**Model 1 ***	**Model 2 ****		
				**RR (95% CI)**	**RR (95% CI)**		
			**Binge drinking**				
			During COVID-19	1.18 (0.97–1.45)	1.27 (1.08–1.48)		
			**Alcohol frequency (drinking 4+ days per week)**				
			During COVID-19	1.06 (0.96–1.17)	1.23 (1.11–1.35)		
			**Heavy drinking (5+ drinks on a typical day when drinking)**				
			During COVID-19	0.60 (0.42–0.86)	0.46 (0.38–0.55)		
			* adjusted for year, age group, gender, ethnicity, period and period × age group interaction				
			** adjusted for year, age group, gender, ethnicity, period and period × gender interaction				
Daly and Robinson, 2021 [80] (Cohort study)	United Kingdom	Adults (follow-up: *n* = 3358)	**Change since lockdown**				
				**2016–2018 (M (SD))**	**May 2020 (M (SD))**	***p***	
			**Overall AUDITPC score**	3.17 (2.46)	3.34 (2.77)	0.003	
			**Change since lockdown**				
				**2016–2018 (%)**	**May 2020 (%)**	***p***	
			**High-risk drinking**	19.3	24.6	0.001	
Alpers et al., 2021 [69] (Cross-sectional study)	Norway	Adults (*n* = 25,708)	**Change since lockdown**				
			**Alcohol consumption**	**Prevalence (*n*%)**			
			Increase	13			
			Decrease	23			
			Association between several risk factors and an increase in alcohol consumption				
				**OR (95% CI) ***			
			**Temporarily lay-off**	1.3 (1.1–1.4)			
			**Quarantine**	1.2 (1.1–1.4)			
			**Home office/study**	1.4 (1.3–1.5)			
			* adjusted for age, gender, economic worries, health worries, temporarily lay-off and/or quarantine and/or home office/study				
Avery et al., 2020 [72] (Cross-sectional study)	United States	Adult twins (*n* = 3971)	**Change since lockdown**				
			**Alcohol consumption**	**Prevalence (%)**			
			Do not use	35.5			
			Use more	14.3			
			Use the same	39.4			
			Use less	10.9			
Cicero et al., 2021 [77] (Cross-sectional study)	Italy	Adults (*n* = 359)	**Change since lockdown**				
				**Pre-quarantine (% (SD))**	**During quarantine (% (SD))**	***p***	
			**Total energy derived from the alcohol**	2.9 (0.6)	4.9 (1.0)	0.002	
Bourion-Bedes et al., 2021 [76] (Cross-sectional study)	France	Students (*n* = 3936)	**Change since lockdown**				
			**Alcohol consumption**	**Prevalence (%)**			
			None	34.2			
			No change	17.1			
			Increased	13.7			
			Reduced	35			
Lechner et al., 2020 [84] (Cross-sectional study)	United States	Students (*n* = 1958)	**Change since lockdown**				
				**Week prior to university closing (M (SD))**	**Week succeeding university closing (M (SD))**		
			**Number of weekly standard drinks**	3.48 (5.45)	5.01 (6.86)		
			**Number of drinking days**	1.36 (1.55)	1.94 (1.84)		
White et al., 2021 [97] (Cross-sectional study)	United States	Students (*n* = 297)	**Change since lockdown**				
				**Pre-closure (M)**	**Post-closure (M)**	***p***	**d**
			**Drinking frequency (in days)**	3	3.2	<0.05	0.12
			**Weekly quantity (drinks/week)**	11.5	9.9	<0.01	0.15
			**Maximum number of drinks in one day**	4.9	3.3	<0.001	0.47

#### 3.4.4. Weight and Body-Mass-Index

In a cohort study, young adults gained around 3.5 pounds on average during COVID-19 restrictions in the US compared to baseline values from October 2018 to October 2019 [85].

**Table 7 ijerph-18-08567-t007:** Results on weight and body mass index.

Reference (Study Design)	Country	Population (Sample Size)	Results			
Adults						
Mason et al., 2020 [85] (Cohort study)	United States	Young adults (baseline: 2013: *n* = 4100, 2020: *n* = 2548, follow-up: 1820)	**Change since lockdown**			
				**M (SD)**	**M% (SD)**	
			Weight change (pounds)	3.47 (14.57)	2.5 % (8.6 %)	
Cicero et al., 2021 [77] (Cross-sectional study)	Italy	Adults (*n* = 359)	**Change since lockdown**			
				**Pre-quarantine (M (SD))**	**During quarantine (M (SD))**	***p***
			Body mass index	26.6 (4.7)	26.9 (4.5)	0.361
Radwan et al., 2021 [90] (Cross-sectional study)	United Arab Emirates	Adults (*n* = 2060)	**Change since lockdown**			
			**Weight**	**Prevalence (*n* (%))**		
			Increase	606 (29.4)		
			Decrease	476 (23.1)		
			Same	978 (47.5)		
Barkley et al., 2020 [74] (Cross-sectional study)	United States	Students (*n* = 184)	**Change since campus closure**			
			**Bodyweight (pounds)**	**Pre-campus closure (M (SD))**	**Post-campus closure (M (SD))**	
			Undergraduate students	175.4 (48.4)	176.8 (48.4)	
			Graduate students	163.7 (45.6)	164.5 (45.6)	
			- no statistically significant (*p* ≥ 0.16) main or interaction effects of time for bodyweight			
Özden and Kilic, 2021 [88] (Cross-sectional study)	Turkey	Nursing students (*n* = 1011)	**Change since lockdown**			
			**Weight**	**Prevalence (%)**		
			Increase	46.9		
			Decrease	33.4		
			Same	19.7		
Berard et al., 2021 [75] (Cross-sectional study)	France	Older adults (aged ≥ 50 years) (*n* = 536)	**Change since lockdown**			
				**Prevalence (*n* (%))**		
			**Weight gain**	137 (25.6)		

#### 3.4.5. Eating Behaviour

A cohort study on pupils aged 8–16 years from Spain found a significantly higher value for adherence to the Mediterranean diet during lockdown compared to September–December 2019, and no difference in values for a low adherence to this form of diet, even though 31.4% stated to have worsened their adherence [87].

**Table 8 ijerph-18-08567-t008:** Results on eating behaviour.

Reference (Study Design)	Country	Population (Sample Size)	Results			
Children and Adolescents						
Medrano et al., 2020 [87] (Cohort study)	Spain	School children aged 8–16 years (baseline: *n* = 281, follow-up: *n* = 113)	**Change since lockdown**			
				**T1 (before lockdown)**	**T2 (during lockdown)**	***p***
				**(M SD))**	**(M SD))**	
			**Adherence to the Mediterranean diet**	5.9 (1.8)	6.4 (1.5)	0.018
			**Low adherence to the Mediterranean diet**	86 (81.1)	81 (76.4)	0.476
			**Change since lockdown**			
				**Prevalence (%)**		
			**Worsening of the adherence to the Mediterranean diet**	31.4		
**Adults**						
Cicero et al., 2021 [77] (Cross-sectional study)	Italy	Adults (*n* = 359)	**Change since lockdown**			
				**Pre-quarantine (M (SD))**	**During quarantine (M (SD))**	***p***
			Energy intake	2568 (322)	2739 (442)	<0.001
			Dietary quality index	42.4 (4.1)	37.8 (4.7)	0.011
			**Change since lockdown for total energy derived from the main diet components**			
				**Pre-quarantine (% (SD))**	**During quarantine (% (SD))**	***p***
			**Total carbohydrates**	49.3 (4.6)	52.6 (6.5)	0.048
			**Simple sugars**	3.1 (0.9)	4.6 (1.1)	0.002
			**Total fats**	28.1 (3.2)	31.4 (2.9)	0.047
			**Added fats**	3.9 (1.1)	4.3 (1.2)	0.021
Garre-Olmo et al., 2020 [82] (Cross-sectional study)	Spain	Adults (*n* = 692)	**Change since lockdown**			
				**Prevalence (*n* (%))**		
			**Worsening dietary pattern**	134 (19.4)		
Radwan et al., 2021 [90] (Cross-sectional study)	United Arab Emirates	Adults (*n* = 2060)	**Change since lockdown**			
			Food intake	**Prevalence (*n* (%))**		
			Increase	655 (31.8)		
			Decrease	344 (16.7)		
			Same	1061 (51.5)		
Berard et al., 2021 [75] (Cross-sectional study)	France	Older adults (aged ≥ 50 years) (*n* = 536)	**Change since lockdown**			
				**Prevalence (*n* (%))**		
			**Decreased diet quality**	142 (26.5)		

#### 3.4.6. Smoking

One cohort study from the UK reported a decreased relative risk for current smoking status and regular e-cigarette use among adults when comparing 2017–2019 with the time of the COVID-19 lockdown (relative risk: 0.89 (95% CI: 0.82–0.97 and relative risk: 0.66 (95% CI: 0.48–0.91, respectively) [32].

**Table 9 ijerph-18-08567-t009:** Results on smoking.

Reference (Study Design)	Country	Population (Sample Size)	Results		
Adults					
Niedzwiedz et al., 2020 [32] (Cohort study)	United Kingdom	Adults (baseline: *n* = 27,141, analysed at follow-up: *n* = 9748)	**Association between lockdown and …**		
				**Model 1 ***	**Model 2 ****
				**RR (95 % CI)**	**RR (95 % CI)**
			**Current smoking**		
			During COVID-19	0.80 (0.69–0.93)	0.88 (0.78–0.98)
			**Regular e-cigarette use**		
			During COVID-19	0.68 (0.46–1.01)	0.61 (0.43–0.86)
			* adjusted for year, age group, gender, ethnicity, period and period × age group interaction ** adjusted for year, age group, gender, ethnicity, period and period × gender interaction		
Cicero et al., 2021 [77] (Cross-sectional study)	Italy	Adults (*n* = 359)	**Change since lockdown**		
				**Prevalence (%)**	
			Reduction	2.2	
			Increase	1.7	
Radwan et al., 2021 [90] (Cross-sectional study)	United Arab Emirates	Adults (*n* = 2060)	**Change since lockdown**		
				**Prevalence (*n* (%))**	
			Increase	50 (21.0)	
			Decrease	93 (39.1)	
			Same	95 (39.9)	
Bourion-Bedes et al., 2021 [76] (Cross-sectional study)	France	Students (*n* = 3936)	**Change since lockdown**		
				**Prevalence (%)**	
			None	83.5	
			No change	3	
			Increased	7.2	
			Reduced	6.3	
Berard et al., 2021 [75] (Cross-sectional study)	France	Older adults (aged ≥ 50 years) (*n* = 536)	**Change since lockdown**		
				**Prevalence (*n* (%))**	
			Increased smoking	21 (4.0)	

#### 3.4.7. Antihypertensive/Lipid-Lowering/Hypoglycaemic Medication

There was only one cross-sectional study, but no prospective study on the influence of the COVID-19 lockdown on antihypertensive, lipid-lowering, and/or hypoglycaemic drug treatment among older adults [75].

**Table 10 ijerph-18-08567-t010:** Results on Antihypertensive/lipid-lowering/hypoglycaemic medication.

Reference (Study Design)	Country	Population (Sample Size)	Results	
Adults				
Berard et al., 2021 [75] (Cross-sectional study)	France	Older adults (aged ≥ 50 years) (*n* = 536)	**Change since lockdown**	
				**Prevalence (*n* (%))**
			**Increased antihypertensive, lipid-lowering, or hypoglycaemic drug treatment**	2 (0.37)

## 4. Discussion

### 4.1. Summary of Findings

We identified 32 epidemiological observational studies which used complete or probability sampling to recruit participants and investigated the influence of COVID-19 lockdown and quarantine measures on modifiable cardiovascular risk factors among the general population. Overall, most studies focused on adults, and only a few were centred on children, adolescents, or older adults. We found consistent results for physical activity and sedentary behaviour, showing that physical activity levels decreased and sedentary behaviour increased among all age groups during COVID-19 lockdowns. Only results on student populations regarding physical activity showed mixed findings, with one study showing a decrease [92], one study describing no change [74], and two studies outlining an increase in physical activity [88,98]. Most studies on alcohol consumption among adults and students showed an increased alcohol consumption due to COVID-19 lockdown measures. In regard to body weight, studies on adults reported that at least 25% of respondents gained weight. Findings among students were inconsistent, with one study showing that nearly 50% of students gained weight [88], and two studies showing no significant change of body mass index or bodyweight since the university closure [74,77]. No studies on weight changes among children and adolescents were retrieved that fulfilled our inclusion criteria. With regards to adults, studies showed that during lockdown, dietary quality worsened and the amount of food intake increased. On the contrary, the only study on children and adolescents outlined a significantly higher adherence to the Mediterranean diet during lockdown compared to the time without lockdown in late 2019 [87]. Findings on smoking were somewhat inconsistent across studies, whereby lockdown measures due to COVID-19 seem to have generally little effect on tobacco consumption according to study findings.

### 4.2. Discussion of Findings

Even though we found a wealth of epidemiological observational studies on the association between COVID-19 lockdown measures and modifiable cardiovascular risk factors among the general population, most studies used non-probabilistic sampling methods. We excluded these studies to ensure a higher generalizability of the review results. Nevertheless, since response in some studies was low and non-responder analyses were missing, selection bias and an accompanying limited internal validity may not be ruled out. We also included studies with very specific study populations like members of a sport association, adults undergoing annual physical check-ups using the WeChat app, or residents of a continuing care community, but only, if researchers conducted complete or probability sampling methods. These studies may not have a high external validity, but due to representative recruitment, they have a high internal validity.

The exposure to COVID-19 lockdowns is difficult to compare with one another for several reasons. First, lockdown measures varied a lot from country to country and further varied within a country, from region to region. Second, even within a specific country, regulations changed several times depending on the infectious situation. Finally, COVID-19 lockdowns comprised a great variety of different measures (e.g., social distancing rules, prohibition of gatherings, travel bans, closure of educational and recreational facilities, orders to stay at home, requirements to work from home, quarantine requirement in case of a COVID-19 disease, etc.) and time spans of the measures.

It should be kept in mind that the impact of quarantine and lockdown measures on lifestyle habits, which are known to be cardiovascular risk factors, is also influenced by other aspects—two of which we discuss in the following as they were found in the included studies.

Regarding the impact of lockdown measures on physical activity, three studies investigated whether there was a difference if someone was physically active or not before the pandemic: two studies demonstrate that young adults and students, who were physically active before lockdown declined their physical activity levels, whereas those who were physically inactive beforehand increased their activity levels [74,83]. In contrast, in the study of Tornaghi et al. [95], highly physically active students before campus closures increased their level of physical activity.

When interpreting the findings on the impact of COVID-19 lockdown measures on modifiable cardiovascular risk factors, the influencing role of mental health should not be neglected, which worsened during lockdown according to two reviews [31,100]. Ten of our included studies found that worsened health behaviours during lockdowns (i.e., increased alcohol consumption, reduced physical activity, increased sedentary behaviour, increased smoking, diminished dietary pattern) are associated with mental health problems like depression, anxiety, worries, or stress among adults and students [69,70,72,73,75,76,78,82,84,92].

Since we did not consider studies on cardiovascular diseases, we cannot judge which impact COVID-19 lockdown measures would have on these. It seems unlikely that the time span of the COVID-19 pandemic and accompanied lockdowns was long enough in order to conduct and subsequently publish adequate research on these hard endpoints. Nevertheless, since this rapid review showed that COVID-19 lockdown measures influence some health behaviours which are known modifiable cardiovascular risk factors, the occurrence of forthcoming cardiovascular diseases is likely. However, this association may be illustrated in future studies.

### 4.3. Practical Implications

To minimize COVID-19 lockdown and quarantine-related (cardiovascular) risky health behaviours, several preventive measures are suggested by experts. Since results on prevention studies have not been published yet, it is not possible at the moment to judge whether these measures have an actual preventive effect.

A review by Dixit and Nandakumar [101] concludes that technology and social media-based interventions can be effectively used for health promotion measures like physical activity promotion, dietary intervention, or smoking cessation during the COVID-19 pandemic. Nevertheless, it should be kept in mind that such online interventions may increase social and health inequalities, particularly for socially deprived children, adolescents, and families, who may not have technological equipment or access to the internet [102].

To enhance physical activity among pupils in the case of school closures, it is recommended not only to send home lessons for subjects like math or biology, but also for physical activity. If schools have the capacity, streaming exercise classes by the physical education teachers are recommended [3]. For all other age groups, online videos and mobile-based apps are recommended for the promotion of physical activity [103]. To further prevent physical inactivity and resulting physical and psychological consequences due to a lockdown, indoor and outdoor activities (e.g., aerobic exercise, strength, flexibility-stretching, and balance exercises) on the individual level are suggested [104,105,106,107]. Outdoor activities should comply with local regulations, take place in close proximity to the home of a person and consider current physical distancing measures [2,108].

For preventing harmful alcohol consumption and smoking during lockdowns, it is recommended to provide psychological supportive programmes, which use an interdisciplinary approach [5,34]. Further, stress management could be used in order to avoid alcohol drinking as a coping mechanism during the pandemic [34].

When planning cardiovascular health promotion for pandemic lockdown situations, targeting not only cardiovascular risk factors itself should be the focus of interventions, but targeting mental health should also be a priority due to its influencing role in the association between COVID-19 lockdown measures and health behaviours like smoking, physical activity, alcohol consumption, and nutrition.

Further, it should be taken into account that recent measures for infection protection could increase health inequalities regarding cardiovascular risk factors and diseases. Thus, primarily preventive measures in living environments should be utilized, which compensate or at least decrease the unfavourable effects on the cardiovascular system and which address socially disadvantaged persons in particular.

### 4.4. Strengths and Limitations

This rapid review is the first review that summarizes up-to-date research on the effects of the worldwide COVID-19 lockdown measures on the entirety of modifiable cardiovascular risk factors from census studies or studies with probability sampling.

Only a few prospective studies were included, which were able to demonstrate a corresponding temporal relationship. Most articles were cross-sectional studies, which asked about temporal changes of cardiovascular risk factors since the introduction of lockdown measures. These findings might be affected by recall bias. In order to emphasize temporal findings, we decided only to report the findings of prospective (cohort and secondary data) studies. The results of cross-sectional studies are included in the summary of results tables.

We are convinced that the current immense number of epidemiological studies on this topic is not the end of the story and that many more studies will be published in the future. Thus, this rapid review can only be regarded as a snapshot of the recent research landscape, providing a first glimpse at relevant results. In order to gain a “final” conclusion on the subject, an update of this rapid review should be conducted when COVID-19 lockdown measures are over and enough time has passed for publishing research accordingly. Such a review should be prepared as a systematic review, searching more than two databases to find as many relevant published papers as possible. Another approach for a continuous update would be to conduct a living systematic review to “incorporate relevant new evidence as it becomes available” [109,110], which seems to be important during the COVID-19 pandemic because the pandemic “has led to an explosion of scientific literature” [111].

The review was conducted following a standardized procedure that is based on experiences of the German Competence Network Public Health COVID-19 [55]. Only during data extraction and risk of bias assessment, we went beyond these recommendations: instead of performing these two review stages by one experienced reviewer and checking it on a random basis by a second reviewer, data extraction and critical appraisal of all 32 studies were double-checked. Since only two databases were searched (which is legitimate in a rapid review), we cannot rule out that we missed some other important studies on the topic.

## 5. Conclusions

This rapid review illustrates that there is a wealth of epidemiological observational studies on the impact of COVID-19 lockdown and quarantine measures on modifiable cardiovascular risk factors. However, only in a few of these studies were participants recruited with complete or probability sampling. Most studies utilized non-probabilistic sampling (e.g., advertisements on social media, convenient mailing lists, or application of the snowball technique), which affect the representativeness of a study population. According to the included studies, physical activity decreased and sedentary behaviour increased among all age groups in the general population during COVID-19 lockdown. Further, among adults, alcohol consumption increased, dietary quality worsened and the amount of food intake increased. Some adults reported weight gain during lockdown measures. Studies on children and adolescents were sparse. Even though only studies using complete or probability sampling were considered, most included studies had methodological flaws like cross-sectional design, low response, or usage of invalid outcome measurement instruments, which could have led to a decrease in the internal validity. Thus, prospective cohort studies exhibiting a high response and using validated outcome measurement tools—across all age groups, and especially in children and adolescents—are needed. To date, it is expected that much more research on the topic will be published. Thus, this rapid review is only a snapshot of the recent scientific landscape, and the evidence should be updated to a later point in time.

## Figures and Tables

**Figure 1 ijerph-18-08567-f001:**
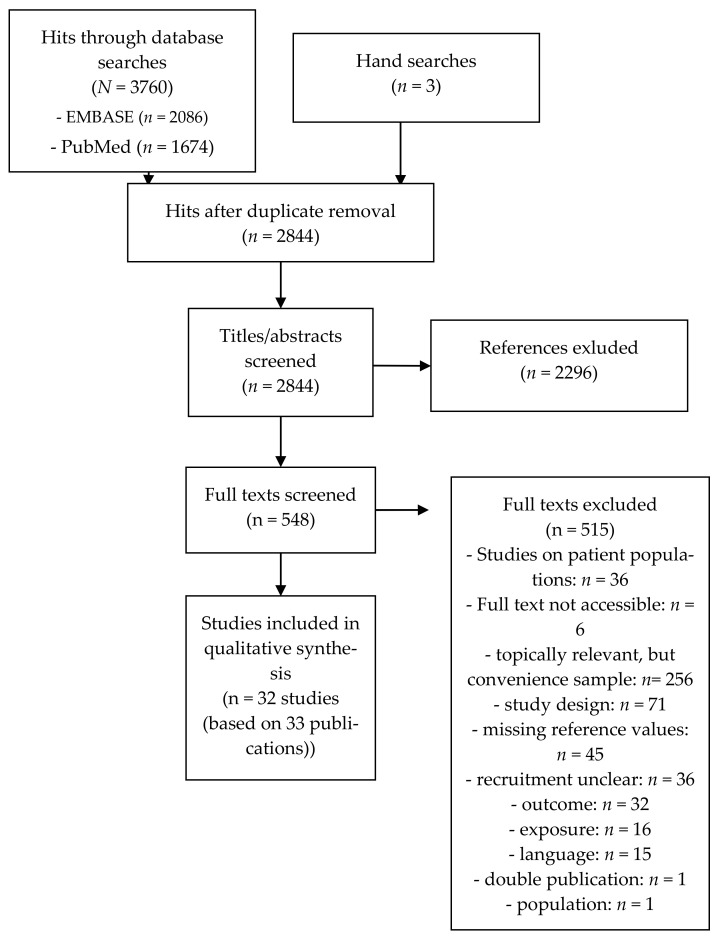
PRISMA-Flow Chart (search date: 17 March 2020).

**Table 1 ijerph-18-08567-t001:** Inclusion and exclusion criteria according to the PECOS-scheme.

Category	Inclusion Criteria	Exclusion Criteria
**Population**	healthy humans of the general population (also including subgroups like pupils, students, or workers) of all ages (i.e., children, adolescents, adults, and older adults)	patient populations only (e.g., obese patients, diabetes patients, patients with cardiovascular diseases) animals
**Exposure**	quarantine/isolation and lockdown measures during the COVID-19 pandemic	quarantine/isolation and lockdown measures during other pandemics (e.g., SARS, MERS, Ebola)
**Comparison**	no or other forms of quarantine/isolation and lockdown measures	no comparison
**Outcome**	modifiable cardiovascular risk factors: physical inactivity sedentary behaviour harmful use of alcohol tobacco use unhealthy diet (excessive consumption of (saturated) fat, salt, and sugar, and low intake of fruits and vegetables) obesity bad blood lipids (hyperlipidaemia, hypercholesterolemia, hypertriglyceridemia) hypertension	non-modifiable cardiovascular risk factors (e.g., family history, diabetes, socioeconomic status) cardiovascular diseases (myocardial infarction, stroke, thrombosis, embolism, arteriosclerosis) other acute or chronic diseases (e.g., mental disorders, cognitive impairments, musculoskeletal disorders) environmental (air pollution, traffic noise) and work-related risk factors (shift work, long working hours)
**Study design**	epidemiological observational studies (cohort studies, case-control studies, cross-sectional studies)	qualitative studies (interview studies, focus group studies) clinical epidemiological studies (case series, case reports) subjective study types (editorial, commentary, expert opinion) animal studies reviews only abstract available

**Table 2 ijerph-18-08567-t002:** Study characteristics of COVID-19 specific primary studies.

Reference, Overall Risk of Bias	Region, Study Design	Time of Survey	Population (Sample Size (% Female), Age (Mean or Median), Response, Lost to Follow-Up (in Cohort Studies))	Exposure *	Outcome
Alpers et al., 2021 [69], High risk	Norway, Cross-sectional study	15–30 April 2020	Adults Sample size: *n* = 25,708 (56.2% female) Age (median (IQR)): 50 years (36–63) Response: 31.7%	COVID-19 pandemic measures (implemented on 12 March 2020) - objectively measures: social distancing, closure of educational, cultural, and training/sport/gym facilities, requirements to work from home, introduction of quarantine requirements - self-reported measures: (a) placed in quarantine, (b) temporarily laid-off, home office/study	Alcohol consumption: Alcohol Use Disorders Identification Test Consumption, self-reported question about change
Anyan et al., 2020 [70], Ernsten and Havnen 2020 [71], High risk	Norway, Cross-sectional study	3–15 June 2020	Physically active adults (members of one Norwegian fitness association) Sample size: *n* = 1314 (30.8% female) Age (mean (SD)): 49 years (11.5) Response: 19.4%	COVID-19 pandemic lockdown (12 March–15 June 2020) - measures: n.r.	Physical activity: self-reported question about change
Avery et al., 2020 [72], High risk	United States, Cross-sectional study	26 March 2020–5 April 2020	(Identical, same-sex fraternal) adult twins Sample size: *n* = 3971 (69.2% female) Age (mean (SD)): 50.4 years (16.0) Response: individual: 32.8%, pairwise: 21.1%	COVID-19 mitigation strategies (Washington implemented the state-wide “stay home, stay healthy” order on 24 March 2020) - measures: n.r.	Alcohol use: self-reported question about change
Barkley et al., 2020 [74], High risk	United States, Cross-sectional study	18 May–18 June 2020	University students Sample size: *n* = 184 (73.2% female (of all participants incl. university staff)) Age (mean (SD)): undergraduate students: 26.9 years (8.9), graduate students: 29.9 years (8.7) Response: 3.7%	Campus closure due to the COVID-19 pandemic (since 11 March 2020) - measures: cancellation of face-to-face classes, closure of the campus, including all fitness facilities, students were sent home, governor’s “stay at home” order (22 March 2020)	Physical activity: Godin physical activity questionnaire Sedentary behaviour: International Physical Activity Questionnaire Weight: self-reported question
Berard et al., 2021 [75], High risk	France, Cross-sectional study	17 April–10 May 2020	Older adults (aged ≥50 years) Sample size: *n* = 536 (52% female) Age (mean (range)): 67 years (50–89) Response: 69%	COVID-19 lockdown (17 March–10 May 2020) - measures: requirement to “stay at home”, prohibition of any gathering of people who did not live in the same home - only reasons for going out: going to work (if teleworking was impossible); doing essential food shopping; traveling for health reasons, assisting vulnerable people, family emergencies, childcare; individual physical activity, taking out a pet (limit: 1 h/day, within a maximum radius of 1 km around the home)	Dietary quality: Short, qualitative food frequency questionnaire Physical activity, weight, smoking, antihypertensive, lipid-lowering or hypoglycaemic drug treatment: self-reported question about change
Bourion-Bedes et al., 2021 [76], High risk	France, Cross-sectional study	7–17 May 2020	Students Sample size: *n* = 3936 (70.6% female) Age (mean (SD)): 21.7 years (4.0) Response: around 7.9%	Lockdown due to the COVID-19 outbreak - measures: n.r.	Alcohol consumption, smoking: self-reported question about change
Cicero et al., 2021 [77], High risk	Italy, Cross-sectional study	n.r.	Adults Sample size: *n* = 359 (56.5% female) Age (mean (SD)): 64.6 years (13.3) Response: 23.3%	COVID-19-related quarantine (February–April 2020) - measures: n.r.	Dietary quality: Dietary Quality Index Alcohol consumption: 1 item from the Dietary Quality Index Smoking, body mass index: 1 self-reported question
Colley et al., 2020 [78], High risk	Canada, Cross-sectional study	29 March–3 April 2020	Adults Sample size: *n* = 4524 (53.4% female) Age: n.r. Response: 62.5%	Physical distancing measures (implemented in March 2020): - measures: border, school, and business closures, avoiding unnecessary trips	Screen time behaviours: 3 self-reported questions
Crochemore-Silva et al., 2020 [79], High risk	Brazil, Cross-sectional study	7–9 May 2020	Adults Sample size: *n* = 377 (62.9% female) Age: n.r. Response: 94.3%	Social distancing - objectively measures: since March 19 adoption of strict social distancing measures (only essential activities in force remained open); 15 and 30 April 2020: suspension of activities in the education network (public and private), social and sports clubs, gyms, cinemas, and bars, amongst others) - self-reported measures: (a) social distancing, (b) activity routine	Leisure-time physical activity: 1 item from an adapted version of the International Physical Activity Questionnaire
Daly and Robinson, 2021 [80], High risk	United Kingdom, Cohort study	T1: 2016–2018 T2: May 2020	Adults born in Britain in 1970 Sample size at follow-up: *n* = 3358 (50% female) Age (range): 46–48 years Response at follow-up: 32.1% Lost to follow-up: n.r.	COVID-19 lockdown restrictions (between late March and early July 2020) - measures: closure of pubs, bars, and restaurants and other nonessential businesses	High-risk alcohol consumption: Alcohol Use Disorders Identification Test
Di Sebastiano et al., 2020 [81], High risk	Canada, (Prospective) secondary data analyses	10 February–19 April 2020 T0: 4 weeks prior physical distancing protocols T1: 1 weeks after the beginning of the physical distancing protocols T2: 6 weeks after physical distancing protocols	Adults (≥18 years) using a physical activity tracking ParticipACTION app Sample size: *n* = 2338 (90.2% female) Age: n.r. Response: n.a. Lost-to follow-up: n.a. (only complete data sets used)	Physical distancing protocols - measures: n.r.	Physical activity: data from a national physical activity tracking app based on steps
Duncan et al., 2020 [73], High risk	United States, Cross-sectional study	26 March–5 April 2020	(Identical, same-sex fraternal) adult twins Sample size: *n* = 3971 (69.2% female) Age (mean (SD)): 50.4 years (16.0) Response: individual: 32.8%, pair-wise: 21.1%	COVID-19 mitigation strategies (Washington implemented the state-wide “stay home, stay healthy” order on 24 March 2020) - measures: n.r.	Physical activity: 1 self-reported question about change
Garre-Olmo et al., 2020 [82], High risk	Spain, Cross-sectional study	8 April–4 May 2020	Adults Sample size: *n* = 692 (54.8% female) Age (mean (SD)): 50.2 years (16.3) Response: 90.5%	Movement restrictions and confinement due to the COVID-19 pandemic (implemented on 15 March 2020) - measures: suspension of all academic activities, obligation to stay at home except to purchase food and medicines, to go to work, or to attend emergencies, more restrictive lockdown period including the temporary closure of all the non-essential activities and businesses (29 March–9 April 2020)	Physical activity, dietary pattern: 1 self-reported question about change
Karuc et al., 2020 [83], High risk	Croatia, Cross-sectional study	24 April–8 May 2020	Young adults Sample size: *n* = 91 (64.8% female) Age (mean (SD)): 21.6 years (0.4) Response: 25.1%	Restrictions due to COVID-19 Pandemic (19 March–11 May 2020) - measures: restriction of gatherings in public places and parks, suspension of public transportation, closing of institutions, prohibition of all social gatherings, work in retail and services including sports activities	Physical activity: 7-day recall of moderate intensity physical activity (MPA) and vigorous intensity physical activity (VPA): School Health Action, Planning, Evaluation System (SHAPES) questionnaire, 1 self-reported question about change
Lechner et al., 2020 [84], High risk	United States, Cross-sectional study	26–31 March 2020	Students (using alcohol in the past 30 days) Sample size: *n* = 1958 (80% female) Age (mean (SD)): 24.94 (7.65) Response: 12.8% (all students)	University closings (on 11 March 2020) - measures: n.r.	Alcohol consumption: Timeline Follow-Back Interview
Mason et al., 2020 [85], High risk	United States, Cohort study	T1: October 2018–October 2019 T2: May–July 2020	Young adults Sample size at follow-up: *n* = 1820 (61.5% female) Age (mean (SD)): 19.72 years (0.47) Response at follow-up: 71.4% Lost to follow-up: n.r.	COVID-19 restrictions - measures: n.r.	Weight: 1 self-reported question about change
McCormack et al., 2020 [86], High risk	Canada, Cross-sectional study	14 April–27 May 2020	Children (5–17 years) Sample size: *n* = 328 (45.1% female) Age: n.r. Response: 4.5% (adults)	COVID-19 public health emergency response - measures: forced closures of educational and day-care facilities, non-essential businesses, and private and public recreation facilities, physical distancing for individuals, forgoing international travel, self-quarantine in case of symptoms	Physical activity, sedentary behaviour: Parents-reported questions about change
Medrano et al., 2020 [87], Low risk	Spain, Cohort study	T1: September–December 2019 T2: March–April 2020	Children (8–16 years) Sample size at follow-up: *n* = 113 Age (mean (SD)): 12.1 years (2.4) Response: 83.6% Lost to follow-up: 61.2	Home confinement during the COVID-19 pandemic - measures: closure of schools, mandatory home confinement for children, total lockdown (children were not allowed to leave their house at all) from 14 March–26 April 2020	Physical activity, screen time: “The Youth Activity Profile” questionnaire Adherence to Mediterranean diet: Mediterranean Diet Quality Index for children and teenagers (KIDMED) questionnaire
Niedzwiedz et al., 2020 [32], Low risk (outcome: “alcohol consumption”), High risk (outcome: “smoking”)	United Kingdom, Cohort study	2015–2020 T1: 2015–2017 T2: 2016–2018 T3: 2017–2019 T4: 24–30 April 2020	Adults (≥18 years) Sample size at follow-up: *n* = 9748 (52.2% female) Age: n.r. Response T4: 48.6% Lost to follow-up T1–T4: 59.6%	COVID-19 lockdown - measures: 12 March 2020: isolation of all with all with symptoms of possible COVID-19 for 7days, 16 March 2020: isolation of all living with someone with symptoms of possible COVID19 for 14 days, advise against unnecessary social contact and travel, banning of mass gatherings, 17 March 2020: advise against all nonessential world-wide travel, 20 March 2020: closure of entertainment, hospitality and indoor leisure premises, schools, colleges and nurseries close for all except children of key workers or children identified as vulnerable, 22 March 2020: advise for extremely clinically vulnerable persons to begin ‘shielding’, 23 March 2020: no permission for the whole population to leave home except for very limited purposes (to buy food; to exercise once per day; for any medical need; to care for a vulnerable person; to travel to/from essential work), banning of all gatherings of more than two people in public, 27 March 2020: public advise to only use open spaces near own house for exercise, and to stay at least 2 m apart from other households while outdoors	Alcohol consumption: Alcohol Use Disorder Identification Test for Consumption: Cigarette smoking: 2 self-reported single questions E-Cigarette use: 1 self-reported question
Okely et al., 2020 [33], High risk	Scotland, Cohort study	T1: 2017–2019 T2: 27 May–8 June 2020	Older adults (born in 1936) Sample size at follow-up: *n* = 137 (48.2% female) Age (mean): 84 years Response: 30.2% Lost to follow-up: n.r.	COVID-19 lockdown (that lasted 34 days at the beginning of data collection): - measures: n.r.	Physical activity: 1 self-reported question
Özden and Kilic, 2021 [88], High risk	Turkey, Cross-sectional study	15–29 May 2020	Nursing students Sample size: *n =* 1011 (60% female) Age (mean (SD)): 19.97 years (3.11) Response: 72.2%	Closure of schools and universities - measures: closure of all schools and universities (16 March 2020), continuation of university education with distance learning possibilities	Weight, exercise: 1 self-reported question about change
Ozturk Eyimaya and Yalçin Irmak, 2020 [89], High risk	Turkey, Cross-sectional study	15–31 May 2020	Children (6–13 years) Sample size: *n =* 1115 (53.4% female) Age (mean (SD)): 9.03 years (1.95) Response: 72.2% (parents)	Lockdown - measures: closure of schools (16 March 2020), temporary lockdown on children and young people (<20 years) (3 April 2020)	Screen time: 1 self-reported question about change
Radwan et al., 2021 [90], High risk	United Arab Emirates, Cross-sectional study	5–18 May 2020	Adults Sample size: *n* = 2060 (75.1% female) Age: n.r. Response: 15.8%	COVID-19 lockdown (from 22 March 2020 onwards) - measures: n.r.	Dietary intake, weight, physical activity, smoking: 1 self-reported question about change
Sasaki et al., 2021 [91], High risk	Japan, Cross-sectional study	August 2020	Older adults (60–95 years) Sample size: *n* = 999 (53.8% female) Age (mean (SD)): 74.5 years (6.3) Response: 74.3%	COVID-19-related distancing restrictions - measures: n.r.	Physical activity: International Physical Activity Questionnaire Short Form Sitting: International Physical Activity Questionnaire Short Form
Savage et al., 2020 [92], High risk	United Kingdom, Cohort study	T1: 14 October 2019 T2: 28 January 2020 T3: 20 March 2020 T4: 27 April 2020	University students Sample size at follow-up: *n* = 214 (72.0% female) Age (mean: 28.0 years Response: 15.6 % Lost to follow-up: 85.5 %	Lockdown: - measures: requirement to stay at home as much as possible, allowance only to leave home once per day for exercise	Physical activity: Exercise Vital Sign (EVS) questionnaire Sedentary behaviour: 1 self-reported question
Schmidt et al., 2020 [93], High risk	Germany, Cohort study	T1: August 2018 T2: 20 April–1 May 2020	Children and adolescents Sample size at follow-up: *n* = 1711 (49.8% female) Age (mean (SD)): 10.36 years (4.04) Response: 25.2% Lost to follow-up: 36.4%	COVID-19 lockdown - measures: closure of kindergartens, schools, sports clubs, gyms, and other leisure institutions relevant to children’s and adolescents organized physical activity (11 March 2020), physical distancing measures and contact restrictions (no more than 2 people from different households to meet in public space), nonorganized sports activities, such as workouts at home, or jogging, and other forms of habitual physical activity besides sports, like going for a walk or playing outside remained allowed if done alone or with people from the same household	Physical activity: MoMo PA Questionnaire Screen time: Self-reported questions
To et al., 2021 [94], High risk	Australia, (Prospective) secondary data analyses	1 January 2018–30 June 2020 (continuous data collection)	Adults (who are registered as members of the 10,000 Steps program) Sample size: *n* = 60,560 (67.0% female) Age: n.r. % active users (of those registered with the app) providing data: 13.1%	Lockdown (2 March 2020) - measures: social distancing guidelines, closure of nonessential businesses, such as gyms, indoor sports facilities, and clubs, allowance to be outside only for exercise or other essential needs, offering of takeaway and delivery services for restaurants and cafes (Relaxation of restrictions: 8 May 2020)	Physical activity: number of steps logged per day (via app)
Tornaghi et al., 2020 [95], High risk	Italy, Cohort study	T1: 27–30 January 2020 T2: 4–10 April 2020 T3: 4–10 May 2020	Adolescents (15–18 years) Sample size at follow-up: *n* = 1568 (% female: n.r.) Age: n.r. Response: 93% Lost to follow-up: 0%	COVID-19 lockdown (11 and 22 March 2020) - measures: abrogation of nonessential movement, including outdoor sports and motor activity, with the exception of activities practiced in a 200 m home-block area and provision of at least 1 m of interhuman distance	Physical activity: International Physical Activity Questionnaire
Wang et al., 2020 [96], High risk	China, Cohort study	T0: 2019 T1: 30 days prior to 21 January 2020 T2: 30 days after 21 January 2020	Middle-aged and older adults Sample size at follow-up: *n* = 3544 (34.6% female) Age (mean (SD)): 51.6 years (8.9) Response: 57.1% Lost to follow-up: 15.0%	Physical distancing measures - measures: n.r.	Walking activity: daily steps collected via a smartphone linked to WeChat
White et al., 2021 [97], High risk	United States, Cross-sectional study	n.r.	College students (who reported drinking alcohol pre- and post-campus closure) Sample size at follow-up: *n* = 297 (62% female) Age (mean (SD)): 21.1 years (0.82) Response: 66%	Campus closure because of COVID-19 - measures: n.r.	Drinking: Daily Drinking Questionnaire
Wickersham et al., 2021 [98], High risk	United Kingdom, (Prospective) secondary data analyses	T1: 23 March 2020 T2: 23 March–10 May 2020 T3: 11 May–14 June 2020 (continuous data collection)	Students (who had enrolled in the remote measurement technology King’s Move Physical Activity tracker app) Sample size: *n* = 763 Age (median (IQR): 22 years (20–25) % active users (of those registered with the app) providing data: 73.5% (but only 2.2% off all students)	COVID-19 lockdown (23 March 2020) - measures: closure of services, including fitness centres, hospitality, leisure, and educational institutions, allowance only go outside for one form of exercise per day or to make essential shopping trips, closure of all university campuses (easing of restrictions: 11 May 2020)	Physical activity: app data (measuring steps walked and miles run per week)
Yamada et al., 2020 [99], High risk	Japan, Cohort study	1 January–25 May 2020(continuous data collection)	Physically independent residents, living in a continuing care retirement community Sample size at follow-up: *n* = 114 Age (range): 67–92 years Response: 38.5% Lost to follow-up: 0%	Social/physical distancing and self-isolation - measures: announcement of the continuing care retirement community of a cancellation of all upcoming in-facility events/exhibitions and the closure of some common facilities as a precaution measure (24 February 2020), state of emergency asking people to stay at home (7 April 2020)	Walking: walking distance within the continuing care retirement community based on behaviour logs from a beacon transmitter

Abbreviations: IQR inter quartile range, n sample size, n.a. not applicable, n.r. not reported, SD standard deviation, T Time of survey. * We use the information provided in the study.

**Table 3 ijerph-18-08567-t003:** Results of the risk of bias -assessment.

Reference	Major Domains	Minor Domains	Overall Risk
1. Recruitment Procedure and Follow-Up (in Cohort Studies)	2. Exposure Definition and Measurement	3. Outcome Source and Validation	4. Confounding and Effect Modification	5. Analysis Method	6. Chronology	7. Blinding of Assessors	8. Funding	9. Conflict of Interest
Alpers et al., 2021 [69] *(for investigation of association between self-reported quarantine status and alcohol consumption)*										
Alpers et al., 2021 [69] *(outcome: change in alcohol consumption)*										
Anyan et al., 2020 [70], Ernsten and Havnen 2020 [71]										
Avery et al., 2020 [72]										
Barkley et al., 2020 [74] *(outcomes: physical activity, sedentary behaviour)*										
Barkley et al., 2020 [74] *(outcome: weight)*										
Berard et al., 2021 [75] *(outcome: dietary quality)*										
Berard et al., 2021 [75] *(outcomes: physical activity, weight, and smoking)*										
Bourion-Bedes et al., 2021 [76]										
Cicero et al., 2021 [77] *(outcome: dietary quality)*										
Cicero et al., 2021 [77] *(outcomes: BMI, smoking)*										
Colley et al., 2020 [78]								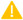	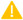	
Crochemore-Silva et al., 2020 [79]								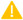	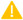	
Daly and Robinson, 2021 [80] a									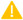	
Di Sebastiano et al., 2020 [81]										
Duncan et al., 2020 [73]										
Garre-Olmo et al., 2020 [82]										
Karuc et al., 2020 [83] *(for investigation of association between quarantine status and physical activity)*										
Karuc et al., 2020 [83] *(outcome: change in physical activity)*										
Lechner et al., 2020 [84]								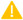		
Mason et al., 2020 [85]										
McCormack et al., 2020 [86]										
Medrano et al., 2020 [87]										
Niedzwiedz et al., 2020 [32] *(Outcome: alcohol consumption)*										
Niedzwiedz et al., 2020 [32] *(Outcome: smoking)*										
Okely et al., 2020 [33]										
Özden and Kilic, 2021 [88]										
Ozturk Eyimaya and Yalçin Irmak, 2020 [89]								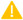		
Radwan et al., 2021 [90]										
Sasaki et al., 2021 [91]										
Savage et al., 2020 [92] *(outcome: physical activity)*										
Savage et al., 2020 [92] *(outcome: sedentary behaviour)*										
Schmidt et al., 2020 [93]										
To et al., 2021 [94]										
Tornaghi et al., 2020 [95]										
Wang et al., 2020 [96]										
White et al., 2021 [97]									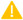	
Wickersham et al., 2021 [98]										
Yamada et al., 2020 [99]								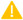	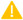	


 Low risk, 

 High risk, 
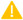
 Unclear risk.

## Data Availability

Not applicable.

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
