# Peer review of "A Rapid Review on the Influence of COVID-19 Lockdown and Quarantine Measures on Modifiable Cardiovascular Risk Factors in the General Population"

_ijerph, 2021, doi:10.3390/ijerph18168567_

Round 1
Reviewer 1 Report
The article summarizes the literature which was published on the topic of its main research objective. It gives in very many details (both in a table and in the text) a description of the methods and results of the reviewed articles, while most of them have 'high risk' of bias. If most articles are not able to demonstrate temporality between the COVID-19 lockdown measures and changing in cardiovascular risk factors, there is no need for so many details. I would assume that those cross-sectional studies had a different purpose than presenting a temporal direction.
As several types of potential bias, in all the reviewed articles, are described, I would suggest adding a table which present a cross-tabulation between the listed articles as its rows and the types of biases as its columns. This will demonstrate all the possible biases in each article. It will help to see the whole picture.
In the results chapter the many details should be summarized as it appears in the beginning of the discussion. By doing so it will enable a more easy reading and will skip repeatability (the discussion may start with 4.2).
Minor remarks:
- Reference no 56 has a wrong link. I could fine the quoted site but not through the link.
- Quoting in the text the names of authors (according to the Vancouver style, which was used here) should not include the names of all authors as is used in this article. For example: in row 355-6)"…but for the prospective cohort study of Niedzwiedz, Green, Benzeval, Camp-355 bell, Craig, Demou, Leyland, Pearce, Thomson, Whitley and Katikireddi [32]" – should be quoted 'Niedzwiedz et al [32]"
Author Response
Responses to Reviewer 1:
Comment 1:
The article summarizes the literature which was published on the topic of its main research objective. It gives in very many details (both in a table and in the text) a description of the methods and results of the reviewed articles, while most of them have 'high risk' of bias. If most articles are not able to demonstrate temporality between the COVID-19 lockdown measures and changing in cardiovascular risk factors, there is no need for so many details. I would assume that those cross-sectional studies had a different purpose than presenting a temporal direction.
Response 1:
Thank you very much for reviewing our manuscript and pointing out several aspects, which help to strengthen the quality of the paper.
The cross-sectional studies on the topic of course did not intend to show a temporality, but wanted to investigate at least if there was a relationship between lockdown and cardiovascular risk factors. But in order to avoid overwhelming the reader with too many written details, we decided – in accordance with your comment – to report descriptively only the results of prospective studies, namely of cohort studies and secondary data analyses. In line with another reviewer’s suggestion, we further included a summary of results table for each risk factor under each appropriate section, which also comprise the results of the cross-sectional studies.
Comment 2:
As several types of potential bias, in all the reviewed articles, are described, I would suggest adding a table which present a cross-tabulation between the listed articles as its rows and the types of biases as its columns. This will demonstrate all the possible biases in each article. It will help to see the whole picture.
Response 2:
Thank you for that hint. We replaced the risk of bias -assessment table, which was initially reported in the Supplement, into the main manuscript (see Table 3).
Comment 3:
In the results chapter the many details should be summarized as it appears in the beginning of the discussion. By doing so it will enable a more easy reading and will skip repeatability (the discussion may start with 4.2).
Response 3:
Thank you very much for this useful hint. To shorten the results chapter, we only report the results of prospective studies in order to show findings on temporal relationships (please refer also to Response 1).
Comment 4:
Reference no 56 has a wrong link. I could fine the quoted site but not through the link.
Response 4:
Do you mean the link for our PROSPERO protocol? I tried the link and it is working on my computer.
Comment 5:
Quoting in the text the names of authors (according to the Vancouver style, which was used here) should not include the names of all authors as is used in this article. For example: in row 355-6)"…but for the prospective cohort study of Niedzwiedz, Green, Benzeval, Camp-355 bell, Craig, Demou, Leyland, Pearce, Thomson, Whitley and Katikireddi [32]" – should be quoted 'Niedzwiedz et al [32]"
Response 5:
Thank you very much for pointing this out. We now changed the citation style in accordance with your recommendation
Reviewer 2 Report
COMMENTS
With respect to the manuscript: A rapid review on the Influence of COVID-19 lockdown and quarantine measures on modifiable cardiovascular risk factors in the general population: Evidence from studies using probability sampling.
A first comment I must make is that the title of the manuscript is too long. I suggest that it be shortened regardless of whether you want to emphasize that it is a quick review. It would not be necessary to remark on it.
A systematic review has several objectives: to evaluate the quality and methodology used in the research conducted in each area of knowledge, to synthesize the scientific evidence and to be useful for decision making.
Regarding the guiding question of the study: “Do COVID-19 lockdown and quarantine measures influence modifiable cardiovascular risk factors in the (healthy) general population of all age groups in comparison to no or other forms of quarantine and lockdown measures?”.
However, ignoring these previous aspects, the guiding question is not very precisely constructed. Reading the context of the study and the proposed question, what we are ultimately seeking to know is the effect of quarantine or confinement on the population with cardiovascular risk factors versus another population without quarantine or periods of confinement.
The question should clarify which is the population and to which age groups we are referring. If you read the question, it is not consistent with the breakdown described by the investigators (PECOS criteria). Even in this description you do not define "S" which refers to research designs. You should align the question with the PECOS criteria you are proposing and if you define the criteria, they should be complete.
The studies evidently, as suggested by the researchers, do not yield more information in this regard. This is complex to analyze due to the periods of interest of the study, both quarantine and confinement periods at different moments in time and with different definitions of variables (remember that over time changes were made in the definition of some variables), which implies different interpretations of the results at different moments in time. Therefore, cross-sectional studies do not allow an adequate analysis of the problem posed by the researchers. However, prospective studies would be more appropriate since some statistical methodologies could better handle these temporal differences to be able to make comparisons for the discussion. Therefore, it would be necessary to locate mainly prospective studies that would allow a better approximation.
Finally, researchers address known risk factors or unhealthy habits that negatively influence the presence of cardiovascular disease. In this case, it would be interesting to know the times of each of the studies, the population and the type of quarantine or confinement used, as well as public policies that were implemented to influence this issue since some countries took measures.
It should be remembered that some effects are not direct and that there is intermediate, interaction and confounding variables.
Perhaps it would be less confounding if we concentrate on the effects in extended quarantine periods that could generate greater effects than those countries where mobile quarantine periods that depended on the evolution of the pandemic were put in place. Thus, some sectors were in quarantine while others were out of quarantine. Interventions were applied in some countries, which in some way could have impacted and modified the effect.
Although the study generated results, I believe that other aspects should be analyzed and that I understand that they are not part of the objective of the study but that they do influence the results and their interpretation. In addition, the guiding question of the study needs to be properly worded so that it can provide an outlet for the results obtained and the conclusions.
Author Response
Responses to Reviewer 2:
Comment 1:
A first comment I must make is that the title of the manuscript is too long. I suggest that it be shortened regardless of whether you want to emphasize that it is a quick review. It would not be necessary to remark on it.
Response 1
Firstly, we would like to thank you very much for your thorough review.
In accordance with your suggestion, we shortened the title as follows: “A rapid review on the Influence of COVID-19 lockdown and quarantine measures on modifiable cardiovascular risk factors in the general population.” We want to keep the information about the review type we used, but we deleted the information on the recruitment procedures.
Comment 2:
A systematic review has several objectives: to evaluate the quality and methodology used in the research conducted in each area of knowledge, to synthesize the scientific evidence and to be useful for decision making.
Response 2:
Thank you very much for pointing this out. As we conducted a rapid review, I am not sure how to proceed with the comment.
Comment 3:
Regarding the guiding question of the study: “Do COVID-19 lockdown and quarantine measures influence modifiable cardiovascular risk factors in the (healthy) general population of all age groups in comparison to no or other forms of quarantine and lockdown measures?”.
However, ignoring these previous aspects, the guiding question is not very precisely constructed. Reading the context of the study and the proposed question, what we are ultimately seeking to know is the effect of quarantine or confinement on the population with cardiovascular risk factors versus another population without quarantine or periods of confinement.
The question should clarify which is the population and to which age groups we are referring. If you read the question, it is not consistent with the breakdown described by the investigators (PECOS criteria). Even in this description you do not define "S" which refers to research designs. You should align the question with the PECOS criteria you are proposing and if you define the criteria, they should be complete.
Response 3:
I am not sure, if I get your point. Most criteria of our PECOS -criteria are formulated within the research question in the same way as it was formulated for the PECOS -criteria.
You write that “we are ultimately seeking to know is the effect of quarantine or confinement on the population with cardiovascular risk factors versus another population without quarantine or periods of confinement”. Of course, that would have been the best way to show the effect of quarantine and confinement. Nevertheless, we were also interested in studies, which for example compared effects in countries with different lockdown measures, in order to show possibly different effects of stricter or milder forms of lockdown rules. Thus, we expanded our definition of the Comparison by including “different forms of quarantine and lockdown measures” besides “no quarantine and lockdown measures”. This criterion of the PECOS -criteria is reported in the research question with the following phrase: “in comparison to no or other forms of quarantine and lockdown measures”.
In regard to the “P” for population, our criteria were healthy persons from the general population of all age groups. This information is part of the research question. To avoid misconception, we rewrote the accordingly phrase (now: “healthy persons from the general population of all ages”). In order to avoid any misunderstanding, we reworded the part about students, pupils, and workers in the section about the detailed PECOS -criteria.
You are right. We did not include the “study design” in the question, which we now made up. Thus, the wording of the question was modified in order to include this PECOS -criteria.
Comment 4:
The studies evidently, as suggested by the researchers, do not yield more information in this regard. This is complex to analyze due to the periods of interest of the study, both quarantine and confinement periods at different moments in time and with different definitions of variables (remember that over time changes were made in the definition of some variables), which implies different interpretations of the results at different moments in time. Therefore, cross-sectional studies do not allow an adequate analysis of the problem posed by the researchers. However, prospective studies would be more appropriate since some statistical methodologies could better handle these temporal differences to be able to make comparisons for the discussion. Therefore, it would be necessary to locate mainly prospective studies that would allow a better approximation.
Response 4:
Thank you very much for this valuable hint. In accordance with your and another reviewer’s comment, we decided only to descriptively report the results of prospective studies. The results of cross-sectional studies are outlined in the summary and data extraction tables. We further added a discussion on this matter in the chapter about “Strength and limitations” of the rapid review.
Comment 5:
Finally, researchers address known risk factors or unhealthy habits that negatively influence the presence of cardiovascular disease. In this case, it would be interesting to know the times of each of the studies, the population and the type of quarantine or confinement used, as well as public policies that were implemented to influence this issue since some countries took measures.
Response 5:
That is right. The information about the time of the survey as well as the population are outlined in Table 2. This table also contains details on the quarantine and confinement measures which were present in each study. Some studies presented comprehensive information on these measures. Others only gave little information.
Comment 6:
It should be remembered that some effects are not direct and that there is intermediate, interaction and confounding variables.
Response 6:
You are right on that. We already included a discussion on this aspect in the discussion about two potential influencing factors, namely physical activity level prior to lockdowns and mental health status. We also took account of the – in our opinion – most important confounding factors – age and gender – by judging these in the course of the risk of bias -assessment.
In order to highlight this situation, we added the following sentence in the discussion: “It should be kept in mind that the impact of quarantine and lockdown measures on lifestyle habits, which are known to be cardiovascular risk factors, is also influenced by other aspects – two of which we discuss in the following as they were found in the included studies.”
Comment 7:
Perhaps it would be less confounding if we concentrate on the effects in extended quarantine periods that could generate greater effects than those countries where mobile quarantine periods that depended on the evolution of the pandemic were put in place. Thus, some sectors were in quarantine while others were out of quarantine. Interventions were applied in some countries, which in some way could have impacted and modified the effect.
Response 7:
Thank you for that thought. We find it difficult to define a strict time frame for lockdown measures which could be seen as extended quarantine periods. Experience within the COVID-19 pandemic shows that quarantine and lockdown measures are very fluid and change rapidly. Measures are strengthened or relaxed within a few days. But it is of course important to discuss this matter, which we already did in the discussion -section.
Comment 8:
Although the study generated results, I believe that other aspects should be analyzed and that I understand that they are not part of the objective of the study but that they do influence the results and their interpretation. In addition, the guiding question of the study needs to be properly worded so that it can provide an outlet for the results obtained and the conclusions.
Response 8:
Again, thank you very much for your opinion. We already reacted to these two aspects in the Responses 3 and 6.
Reviewer 3 Report
This is a timely and comprehensive literature review on the impact of the COVID-19 pandemic and lockdown on life style focusing on cardiovascular risk factors. The manuscript is well organized and the methods for this meta-analysis well described. I have only minor suggestions:
- Please add tables summarizing the main findings in each result section (including sample size, study population, country, relative risk, values of p).
- Please add a figure summarizing relative risks/values of p for each lifestyle factor in 4.1 Summary of findings.
Author Response
Comment 1:
This is a timely and comprehensive literature review on the impact of the COVID-19 pandemic and lockdown on life style focusing on cardiovascular risk factors. The manuscript is well organized and the methods for this meta-analysis well described. I have only minor suggestions:
Response 1:
First of all, thank you very much for taking you precious time for revieing our paper.
Comment 2:
Please add tables summarizing the main findings in each result section (including sample size, study population, country, relative risk, values of p).
Response 2:
Thank you for this suggestion. In accordance with your comment, we added such tables after each section.
Comment 3:
Please add a figure summarizing relative risks/values of p for each lifestyle factor in 4.1 Summary of findings.
Response 3:
As the results of our rapid review varied a lot in regard to outcomes investigated and the appropriate operationalization method used in the studies, we forwent to conduct a meta-analysis, but instead decided to present the results descriptively and tabularly. Due to this heterogeneity, the results are not presentable in one figure, in our opinion.
Round 2
Reviewer 1 Report
Thanks for adding a table of the type of bias in the articles which were reviewed. It visualizes the very many biases in those articles. This presentation should have led the authors to refer in their review only to the articles which have fewer potential biased. The shortening of the results in the text is appreciated. I suggest to skip all the other details, by skipping tables 4-10.
Author Response
Thank you again for your review. As the implementation of Tables 4-10 was a requirement by another reviewer we do not want to delete these tables.
Reviewer 2 Report
Regarding the manuscript in its second version: A rapid review on the influence of COVID-19 lockdown and quarantine measures on modifiable cardiovascular risk factors in the general population.
I believe that the authors have applied most of the suggestions that were indicated and that have been highlighted by the researchers.
For my part, I have no additional observations. I believe that its publication can be approved.

Author Response
Thank you very much again for your review. Best regards.